# Anti-*Pseudomonas aeruginosa* Vaccines and Therapies: An Assessment of Clinical Trials

**DOI:** 10.3390/microorganisms11040916

**Published:** 2023-03-31

**Authors:** Moamen M. Elmassry, Jane A. Colmer-Hamood, Jonathan Kopel, Michael J. San Francisco, Abdul N. Hamood

**Affiliations:** 1Department of Biological Sciences, Texas Tech University, Lubbock, TX 79409, USA; 2Department of Medical Education, Texas Tech University Health Sciences Center, Lubbock, TX 79430, USA; 3Department of Immunology and Molecular Microbiology, Texas Tech University Health Sciences Center, Lubbock, TX 79430, USA; 4Department of Medicine, Texas Tech University Health Sciences Center, Lubbock, TX 79430, USA; 5Honors College, Texas Tech University, Lubbock, TX 79409, USA; 6Department of Surgery, Texas Tech University Health Sciences Center, Lubbock, TX 79430, USA

**Keywords:** ventilator-associated pneumonia, clinical trials, vaccines, cystic fibrosis, chronic lung infection, antibiotics, immunotherapy, bacteriophages, *Pseudomonas aeruginosa* virulence factors, biofilms

## Abstract

*Pseudomonas aeruginosa* is a Gram-negative opportunistic pathogen that causes high morbidity and mortality in cystic fibrosis (CF) and immunocompromised patients, including patients with ventilator-associated pneumonia (VAP), severely burned patients, and patients with surgical wounds. Due to the intrinsic and extrinsic antibiotic resistance mechanisms, the ability to produce several cell-associated and extracellular virulence factors, and the capacity to adapt to several environmental conditions, eradicating *P. aeruginosa* within infected patients is difficult. *Pseudomonas aeruginosa* is one of the six multi-drug-resistant pathogens (ESKAPE) considered by the World Health Organization (WHO) as an entire group for which the development of novel antibiotics is urgently needed. In the United States (US) and within the last several years, *P. aeruginosa* caused 27% of deaths and approximately USD 767 million annually in health-care costs. Several *P. aeruginosa* therapies, including new antimicrobial agents, derivatives of existing antibiotics, novel antimicrobial agents such as bacteriophages and their chelators, potential vaccines targeting specific virulence factors, and immunotherapies have been developed. Within the last 2–3 decades, the efficacy of these different treatments was tested in clinical and preclinical trials. Despite these trials, no *P. aeruginosa* treatment is currently approved or available. In this review, we examined several of these clinicals, specifically those designed to combat *P. aeruginosa* infections in CF patients, patients with *P. aeruginosa* VAP, and *P. aeruginosa*–infected burn patients.

## 1. Introduction

*Pseudomonas aeruginosa* is one of the most prevalent pathogens leading to acute and chronic infections in wounds, lung, and urinary tract [1,2]. *Pseudomonas* species were the third most common cause of Gram-negative infections and accounted for 4% of cases in a prospective analysis of the SCOPE (Surveillance and Control of Pathogens of Epidemiologic Importance) database of 24,179 hospital-acquired infections that occurred in 49 hospitals in the United States between 1995 and 2002 [3]. In recent years, there has been an increase in *P. aeruginosa* infection rates among hospitalized patients, particularly in hospital-acquired infections (Figure 1), lower respiratory tract infection (Figure 2), and bloodstream infection (Figure 3). Furthermore, the rise of multi-drug-resistant *P. aeruginosa* species poses a morbidity and mortality risk on a global scale [4]. The World Health Organization classifies *P. aeruginosa*–resistant bacteria as critical-priority microorganisms (WHO). In fact, the combined *P. aeruginosa’s* resistance, multifactorial pathogenicity, and capacity for overadaptation make it particularly challenging to eradicate from infected patients. This review focuses on new treatments for *P. aeruginosa* infections that are presently undergoing human testing and could be made available to patients. We organized the current review as follows: First, we described the three main types of *P. aeruginosa* infections (ventilator-associated pneumonia (VAP), cystic fibrosis (CF) infection, and burn wound infection) that include the epidemiology, pathophysiology, and clinical presentation; second, we then described the different *P. aeruginosa* virulence factors that play a critical role in the above-described infections; lastly, we examined the different treatments/vaccines/immunotherapies that were used or are currently used in clinical trials.

### 1.1. Ventilator-Associated Pneumonia due to P. aeruginosa Infection

Ventilator-associated pneumonia (VAP) is one of the most common infections reported in patients admitted to the intensive care units (ICUs). The most common infection identified is VAP in those with multi-drug-resistant organisms, particularly *P. aeruginosa* [64,65,66]. The biggest risk factors for *P. aeruginosa* patients with VAP include being placed on a mechanical ventilator longer than 5 days and previous antibiotic exposure [67,68,69]. Patients with chronic obstructive pulmonary disease (COPD) and other chronic respiratory disorders are at risk of developing a serious respiratory infection [67,68,69]. In intubated patients with a history of pneumonia as well as during the postoperative phase following lung transplantation, *P. aeruginosa* is the primary cause of pneumonia [70]. In addition, the most frequent pathogen isolated from individuals with health-care-associated pneumonia who needed ICU admission and mechanical ventilation is *P. aeruginosa* [71].

Despite clear risk factors for VAP, the treatment for VAP remains controversial. The original guidelines for treating VAP called for an antipseudomonal cephalosporin (cefepime, ceftazidime), carbapenem with antipseudomonal fluoroquinolone, or aminoglycoside [72]. However, the guidelines have changed with new clinical data indicating that this treatment regimen often used insufficient dosing, which led to increased morbidity, mortality, and recurrence of *P. aeruginosa* infections [72]. In various trials and meta-analyses, empiric combination therapy with a beta-lactam and an aminoglycoside was shown to be superior to monotherapy in the treatment of *P. aeruginosa* VAP, lowering mortality by up to 50% [73,74,75]. However, there is no difference between using one or two effective antibiotics; consequently, once microbiological data are obtained, monotherapy should be used instead of dual treatment [75]. The antibiotic of choice and the duration are also important considerations when treating *P. aeruginosa* VAP to prevent the emergence of multi-drug-resistant strains and reduce drug toxicity from prolonged antibiotic use [73,74,75]. With the rise in multi-drug-resistant strains, further research is being undertaken to address the need for safer and more effective antibiotics as well as vaccinations against *P. aeruginosa* VAP.

### 1.2. Cystic Fibrosis (CF) and Pneumonia Secondary to P. aeruginosa Infection

Cystic fibrosis (CF) is the most prevalent autosomal recessive genetic condition that is caused by a defect in the cystic fibrosis transmembrane conductance regulator (CFTR) protein and forms a chloride and bicarbonate channel [76,77,78]. The CFTR protein controls the ion transport and, ultimately, the hydration of the respiratory epithelial cells [79,80]. Mutations in CFTR, as seen in CF patients, cause dry and thick mucus discharges that affect multiple organ systems, including the hepatobiliary system, pancreas, gastrointestinal tract, and male reproductive tract [79,80]. The major cause of morbidity and death among CF patients is lung infection. Due to inadequate mucociliary clearance, the airways become vulnerable to bacterial and other infections, many of which develop into chronic and lifelong conditions [81]. With aging, the microbiological environment of CF lung tissue changes [81]. Younger pediatric patients are more likely to contract *Haemophilus influenzae* and *Staphylococcus aureus*, particularly methicillin-sensitive *S. aureus* and more recently methicillin-resistant strains [81]. Additional infections with built-in resistance to numerous antibiotics, such as *Stenotrophomonas maltophilia, Achromobacter xylosoxidans, Burkholderia cepacia* complex and nontuberculous mycobacteria, may predominate with advancing age or illness [82,83]. *Pseudomonas aeruginosa* often becomes the main bacteria cultured from the respiratory tract by adulthood. The incidence of *P. aeruginosa* infection in CF patients increases from 20% in individuals under the age of 5 to as high as 70% by the age of 18 [84,85]. Chronic lung infection with *P. aeruginosa* causes a rapid decline in pulmonary function that contributes to the death of many CF patients [84,85]. Thus, treatment of *P. aeruginosa* infection has become an integral part of CF care.

*Pseudomonas aeruginosa* isolates have unique characteristics that aid in their adaptability to and survival in the CF lung. The main one of these characteristics is the hypermutable genetic background, which leads to the production of *P. aeruginosa* species with distinct variations [86]. These variations include a mucoid phenotype, altered antigenic structures, and antibiotic resistance [86]. Extracellular alginate synthesis, which results in the mucoid phenotype, might lessen the efficacy of antibiotics by restricting drug penetration [86]. Additionally, available evidence suggests that *P. aeruginosa* forms biofilms in the lower respiratory system of *P. aeruginosa* infections in CF patients, making the organism immune to a variety of antibiotics [87,88]. High intrapulmonary drug concentrations can be achieved using aerosolized antibiotics with less systemic adverse effects. This treatment lessens inflammation and reduces the emergence of multi-drug-resistant *P. aeruginosa* density while maintaining lung function and reducing the frequency of acute pulmonary exacerbations [89,90]. Despite the wide variety of treatments for CF, a sizable portion of patients (25%) do not completely regain the lost lung function [91,92]. Therefore, there is a great need for more effective treatments to help reduce *P. aeruginosa*.

### 1.3. Pseudomonas aeruginosa Burn Wound Infections

The most frequent cause of morbidity and mortality in burn patients is still bacterial infection. Due to the numerous physiologic characteristics that make burn damage distinctive, the diagnosis and treatment of burn wound infections continues to be difficult [93]. Patients who experience a serious burn are at a high risk of having burn wound sepsis. Burn wound infection or sepsis is evident through an abrupt change in the burn wound’s appearance or the burn patient’s clinical status [93]. Several types of burn wound infection are classified based on the clinical characteristics and the level of invasion, which are assessed by analyzing the cultures and the histology of burn wound biopsies [93]. Although *Staphylococcus* and *Pseudomonas* are still the most common bacteria causing infections in burn wounds, the epidemiology of burn wound infections has evolved over time [93]. To properly treat burn wound infections, it is critical to be informed of the microbial flora and the antibiotic susceptibility in each burn unit and infected patient [93]. Depending on the kind of burn wound, a combination of wound cleaning, debridement, and topical or systemic antimicrobial medication is used to treat burn wound infection/sepsis [93].

In recent years, *P. aeruginosa* infections obtained from infected burn wounds were found to be resistant to several classes of antibiotics. For example, 20% of the *P. aeruginosa* isolates obtained from infected burn patients were resistant to meropenem, 76% were resistant to gentamicin and imipenem, and 89% were resistant to ticarcillin–clavulanate [94]. In another study, 71% of *P. aeruginosa* isolates were multidrug resistant [95]. In accordance with another study from a burn hospital in Sweden, 26% of *P. aeruginosa* isolates between 1994 and 2012 were carbapenem resistant [96]. Due to the emergence of multi-drug-resistant *P. aeruginosa* species, a combination of two antibiotics is recommended in burn patients with severe *P. aeruginosa* wound infections.

Several *P. aeruginosa* virulence factors dispersed into the environment or injected into host cells or other bacteria are responsible for *P. aeruginosa*’s pathogenesis [97]. These virulence factors alter or impair the signaling pathways of host cells, target the extracellular matrix, cause tissue damage, and allow *P. aeruginosa* to successfully compete with other pathogens and alter the local microbiota. *P. aeruginosa* virulence is combinatorial and multifaceted. Among the several well-characterized antibiotic resistance mechanisms in *P. aeruginosa* include the development of antibiotic-inactivating enzymes, intrinsic membrane permeability, drug efflux systems, loss of porin function, and drug efflux system [97]. The plasticity of the *P. aeruginosa* virulence factor gene expression, antibiotic resistance, and metabolism in response to selective pressure contributes to the ability *of P. aeruginosa* to shift its infection from acute to chronic. In the following section, we describe different *P. aeruginosa* cell–associated and extracellular virulence factors and their role in *P. aeruginosa* infections.

## 2. *P. aeruginosa* Pathogenesis and Virulence Factors

The ability of *P. aeruginosa* to successfully cause different types of infections is related to the bacteria’s large genome (5–7 megabases), which harbors a variety of virulence factors [98]. These virulence factors (i.e., type IV pili, flagella, exopolysaccharides (EPS), lipopolysaccharide (LPS), pyocyanin, siderophores, and secretion systems) serve specific functions in *P. aeruginosa* pathogenesis during infection. Many of these virulent factors are controlled by a complex regulatory network. This network is tightly regulated by a cell-to-cell communication system called quorum sensing (QS) [99]. *P. aeruginosa* has three well-studied QS systems, *las*, *rhl*, and *pqs* [100,101,102]. Type IV pili and flagella are important for *P. aeruginosa* in cell motility, adhesion, and colonization [103,104]. Different strains of *P. aeruginosa* produce three exopolysaccharides (EPS): Pel, Psl, and alginate. These proteins aid in cell adhesion and protect *P. aeruginosa* from the human immune cells [105,106]. Alginate is an overproduced EPS by mucoid strains of *P. aeruginosa*, which is a common characteristic of isolates from chronic lung infections [107,108]. LPS is a major part of the outer membrane of Gram-negative bacteria. It consists of three components: a membrane-anchored lipid A, a core oligosaccharide, and a highly variable O-antigen [109]. The components of LPS are necessary for cell motility, adhesion, and both eliciting and evading the immune system [110]. Among the secreted molecules, pyocyanin is a cytotoxic redox-active metabolite, while siderophores, such as pyoverdine, chelate iron for uptake into *P. aeruginosa* [111,112].

*P. aeruginosa* also has several secretion systems that are used to translocate proteins, known as effectors, to damage the host. Specifically, the type I secretion system of *P. aeruginosa* secretes a few virulence factors, such as an alkaline protease, that interfere with the activation of the complement system of the host immune system [113]. The type II secretion system secretes the largest number of proteins as virulence factors including elastases (LasB and LasA), protease IV, and exotoxin A [114]. LasB, LasA, and protease IV are important proteases in damaging host tissues [115,116]. LasB is an elastolytic metalloprotease associated with *P. aeruginosa* infections as an important virulence factor. This is due to its ability to degrade numerous constituents of the host tissue and immune system [117]. LasB has been also linked to biofilm formation [118,119]. LasA is a staphylolytic metallopeptidase with elastolytic activity [120]. It also enhances the elastolytic activity of LasB [121]. Protease IV is an important virulence factor because it alters immune response by degrading cytokines, such as IL-22 [122]. Exotoxin A is an adenosine diphosphate (ADP)–ribosylating toxin that inhibits the host elongation factor 2. This disrupts protein synthesis, leading to host cell death [123]. Exotoxin A also interferes with host immune response to *P. aeruginosa* infection [124]. In contrast with the type I and II secretion systems, *P. aeruginosa* uses the type III secretion system to translocate effectors, such as exoenzyme S (ExoS), exoenzyme T (ExoT), exoenzyme U (ExoU), and exoenzyme Y (ExoY), directly into the host cell [114,125]. ExoS inhibits phagocytosis and enhances the dissemination of *P. aeruginosa* during pneumonia [126,127]. ExoT is another effector that induces apoptosis in its target cell, while ExoU is another important virulence factor that is critical to *P. aeruginosa* pathogenesis during pneumonia [128,129]. In contrast, ExoY disrupts the endothelial barrier and enhances lung infiltration [130]. Lastly, the type VI secretion system is the most recently identified secretion system in *P. aeruginosa* that delivers several toxins that attack other pathogens as well as the host [131].

One major factor that contributes to the pathogenesis of *P. aeruginosa* within the bloodstream is its serum resistance. Some of the potentially involved cellular components in *P. aeruginosa* serum resistance include alterations in lipopolysaccharide components (e.g., O-antigen) and outer membrane proteins [132,133,134]. Several specific proteins were found to contribute to serum resistance, such as VacJ (involved in maintaining outer membrane lipid asymmetry), AmpD (beta-lactamase expression regulator), MexR (multidrug resistance operon repressor), OprD (outer membrane porin), HepP (a heparinase and a virulence factor), Wzz (necessary for LPS biosynthesis), and WaaL (necessary for LPS biosynthesis) [135,136,137,138]. Several of those identified virulence factors have been proposed as therapeutic targets against *P. aeruginosa;* however, further clinical research is needed to evaluate their efficacy. These identified virulence factors have been extensively studied in lab models prior to clinical trials.

## 3. Clinical Trials to Assess the Effectiveness of Anti-*Pseudomonas aeruginosa* Treatments

As it develops into a chronic stage, *P. aeruginosa* is extremely challenging to treat. Along with our understanding of the molecular processes that underlie *P. aeruginosa*, the different treatment algorithms and the list of potential targets for treatments or vaccine development are growing [139,140,141,142,143,144,145,146,147,148,149,150,151,152,153,154,155,156,157,158,159,160,161,162,163,164,165,166,167,168,169,170,171,172,173,174,175,176,177,178,179,180,181,182,183,184,185,186]. While many of the discoveries made in murine models have not yet been translated into clinical research, several clinical trials were conducted to investigate the translation of information learned from the murine models into treatment or prevention of human *P. aeruginosa* infections. Information on these clinical trials, discussed in more detail below, was obtained from the ClinicalTrials database [187]. We organized different clinical trials as follows: new antibiotics, bacteriophages, strategies targeting *P. aeruginosa* virulence (biofilm, quorum sensing, type III secretion system, and antimicrobial peptides), immunotherapy, outer membrane proteins as a vaccine, strategies targeting *P. aeruginosa* iron acquisition systems.

### 3.1. Antibiotics

#### 3.1.1. New Antibiotics

Given the rise of *P. aeruginosa* multi-drug-resistant strains, there is an interest in developing new antimicrobial drugs using combinations of older medications or novel agents (Table 1A). A broad-spectrum combination antibiotic composed of tobramycin and fosfomycin (FTI) was investigated in a phase II trial to treat CF patients with persistent *P. aeruginosa* infections (**NCT00794586**) [188]. The inhaled FTI combination consisted of two antibiotics: tobramycin, an aminoglycoside antibiotic with a strong activity against Gram-negative pathogens, and fosfomycin, a broad-spectrum combination antibiotic with activity against both Gram-positive and Gram-negative bacteria [188]. Based on clinical studies that examined clinical isolates of *P. aeruginosa*, the concentration of tobramycin and fosfomycin together (4:1 (*w*/*w*) fosfomycin/tobramycin combination) was found to be the same or lower than the inhaled tobramycin alone [188,189]. For example, *P. aeruginosa* isolates from non-CF patients that were treated with the FTI combination versus tobramycin or fosfomycin had a minimum inhibitory concentration at 50% (MIC_50_) of 4 mg/dL (FTI), 1 mg/dL (tobramycin), and 32 mg/dL (fosfomycin) [189]. When the same experiment was repeated to determine the minimum inhibitory concentration at 90% (MIC_90_) for the FTI combination versus tobramycin or fosfomycin, the MIC_90_ for all three groups was the same [189]. A similar trend was observed for both the MIC_50_ and MIC_90_ when the FTI combination versus tobramycin or fosfomycin was tested against *P. aeruginosa* isolates from CF patients. The efficacy of the FTI combination and the tobramycin had similar efficacy, which suggests that the FTI combination may minimize long-term toxicity from repeated exposure to aminoglycosides such as tobramycin, which can cause nephrotoxicity and ototoxicity [188,189]. Therefore, a recent clinical trial assessed the safety and effectiveness of two dose combinations of fosfomycin/tobramycin (FTI) for inhalation in individuals with cystic fibrosis and *P. aeruginosa* lung infection [188]. The clinical trial had 120 adult CF patients with forced expiratory volume in 1 s (FEV1) greater than or equal to 25% and less than or equal to 75% predicted split into the control and treatment groups [188]. The treatment group consisted of two subgroups that received 80 mg/20 mg or 160 mg/40 mg inhaled FTI twice daily for 28 days [188]. The control group received an aztreonam for inhalation solution (AZLI) placebo inhaler twice daily for 28 days [188]. After 28 days, the relative change in lung function from baseline after 28 days was measured for both the treatment and control groups.

Another phase I clinical trial investigated the efficacy of fosfomycin IV (ZTI-01) against persistent *P. aeruginosa* infections (**NCT02178254**) [190]. ZTI-01 inhibits peptidoglycan assembly, thereby disrupting cell wall synthesis [190]. The intravenous form of fosfomycin is thought to be superior to oral dosing for CF patients infected with *P. aeruginosa* [190]. The study included 30 healthy individuals between the ages of 18 and 45 who were randomly assigned to one of three treatment sequences that lasted between 18 and 26 days [190]. The study was designed to determine the safety, tolerability, and pharmacokinetics of two single doses of ZTI-01 (1 and 8 g infused over 1 h) and a single dose of the reference label drug, Monurol^®^ (oral sachet, 3 g) [190]. The first treatment sequence consisted of 10 patients receiving 1.0 g of intravenous ZTI-01 for period 1 (1 h infusion), 8.0 g IV ZTI-01 for period 2 (1 h infusion), and 3 g oral sachet of Monurol in period 3 [190]. For the second treatment sequence, 10 patients received 8.0 g IV ZTI-01 for period 1 (1 h infusion), 3 g oral sachet of Monurol in period 2, and 1.0 g of IV ZTI-01 for period 3 (1 h infusion) [190]. In the third treatment sequence, 10 patients received 3 g oral sachet of Monurol in period 1, 1.0 g of intravenous (IV) ZTI-01 for period 2 (1 h infusion), and 8.0 g IV ZTI-01 for period 3 [190].

One of the newly discovered antibiotics is murepavadin, which interferes with LPS transport in Gram-negative bacteria. Murepavadin (POL7080) belongs to a novel class of antibiotics, known as outer membrane protein targeting antibiotics (ompTAs) [191,210]. It was discovered through screening a large library of peptidomimetic macrocycles that were based on a truncated structure of the antimicrobial peptide protegrin I (PG-1) [211]. Protegrin-1 is a small peptide that contains 18 amino acid residences found in bacteria and fungi [211,212,213]. The structure of PG-1 contains six positively charged arginine residues and four positively charged cysteine residues that form two antiparallel β-sheets with a β-turn [211,212,213]. Murepavadin binds to the lipopolysaccharide transport protein D (LptD), an outer membrane protein involved in lipopolysaccharide biogenesis in Gram-negative bacteria. By binding to LptD, murepavadin inhibits the LPS transport function of LptD and causes lipopolysaccharide alterations in the outer membrane of the bacterium and, ultimately, cell death [210,211]. In vitro analysis demonstrated the strong bactericidal effects of murepavadin against 1219 *P. aeruginosa* isolates (many multi-drug-resistant) obtained from 112 medical clinical centers in the US, China, and Europe. These studies estimated the MIC_50_ of murepavadin against several *P. aeruginosa* isolates of 0.12 mg/liter [214].

Given its effectiveness in preclinical studies, a clinical trial (**NCT02096315**) was conducted to test whether murepavadin is effective in patients with exacerbation of non–cystic fibrosis bronchiectasis caused by *P. aeruginosa* infection [191]. The study administered murepavadin to 20 patients between 18 and 80 years of age with non–cystic fibrosis bronchiectasis caused by *P. aeruginosa* infection for a period of 20 days [191]. The primary outcome of the study was the reduction in CFU/mL (colony-forming units/mL) of *P. aeruginosa* [191].

Polyphor Ltd. followed up on this study with another clinical trial (**NCT02110459**) to investigate the pharmacokinetics and safety of a single dose of murepavadin in patients with mild, moderate, severe, and end-stage renal disease after a single intravenous infusion of murepavadin [192]. The study included 32 adults between ages 18 and 79 divided into four groups with either mild (group 1), moderate (group 2), and severe (group 3) renal function impairment or with normal renal function (group 4) [192]. All patients in the study received a single 2.2-mg/kg of body weight intravenous infusion of murepavadin administered over 3 h [192]. The study found that murepavadin was 2.0- to 2.5-fold higher in patients with renal function impairment compared with patients with normal renal function. Murepavadin was well tolerated in all groups. The study concluded that murepavadin was well tolerated by patients with renal disease with transient and mild adverse side effects [192].

Polyphor Ltd. conducted two clinical trials to evaluate the effectiveness of murepavadin (PRISM-MDR (ventilator-associated pneumonia) and PRISM-UDR (nosocomial pneumonia); **NCT03409679 and NCT03582007, respectively**). PRISM-MDR (**NCT03409679**) was a phase III randomized clinical trial that was designed to investigate the efficacy, stability, and tolerability as well as the pharmacokinetics of the intravenous injection of murepavadin with one of the other antipseudomonal antibiotics (piperacillin/tazobactam, ceftazidine, cefepime, meropenem, amikacin, ciprofloxacin, levofloxacin, colistin) to treat ventilator-associated bacterial (VAB) pneumonia [193]. The study included a total of 41 patients who developed VAB within 48 h after intubation to assess whether murepavadin increased the clinical cure rate 21–24 days after the start of treatment [193]. Patients in the study received either murepavadin every 8 h and one antipseudomonal antibiotic (piperacillin/tazobactam, ceftazidine, cefepime, meropenem, amikacin, ciprofloxacin, levofloxacin, colistin) and two antipseudomonal antibiotics (piperacillin/tazobactam, ceftazidine, cefepime, meropenem, amikacin, ciprofloxacin, levofloxacin, colistin) [193].

PRISM-UDR (**NCT03582007**) is a phase III, multicenter, open-labeled study to investigate the efficacy, stability, and tolerability of intravenous murepavadin given with ertapenem (beta-lactam antibiotic) versus one antipseudomonal antibiotic (meropenem or piperacillin/tazobactam) antibiotic in the treatment of nosocomial pneumonia [194]. The study included two patients who would receive either murepavadin and ertapenem or one antipseudomonal antibiotic (meropenem or piperacillin/tazobactam) to assess the all-cause mortality rates 28 days after the start of study treatment [194]. However, Polyphor Ltd. halted the enrollment due to the higher-than-expected acute kidney injury incidence (56%) observed in the murepavadin arm of the PRISM-UDR trial [194]. Investigators suggested that in patients with moderate or severely impaired renal functions, murepavadin’s dosage should be adjusted accordingly [194].

Another investigative drug is RC01, which, like murepavadin, functions by targeting LPS [195]. A phase I randomized clinical trial (**NCT03832517**) was designed to assess the safety, tolerability, and pharmacokinetics of RC01 in eight healthy volunteers. In the first part of the study, healthy patients received a single-dose escalation of increasing intravenous doses of RC-01. In the second part of the study, healthy patients received multiple-dose escalation of increasing intravenous doses of RC01 given either twice daily or three times daily.

#### 3.1.2. Polymyxin B Derivatives

Despite their propensity for human nephrotoxicity and neurotoxicity, polymyxins are still a family of antibiotics that are available to treat infections caused by multidrug Gram-negative bacteria almost 60 years after their clinical approval [215]. Polymyxins are small cyclic cationic lipopeptides that interact with the anionic lipid A component of LPS in the Gram-negative bacteria’s outer membrane, causing disruption of the cytoplasmic membrane and bacterial cytotoxicity [215]. In recent years, the clinical usage of polymyxin B and polymyxin E has resumed. Many attempts have been made to alter the structure of polymyxins and enhance their safety profile. A polymyxin B derivative with lower nephrotoxicity is SPR741. SPR741 does not directly kill bacteria, but it increases the effectiveness of coadministered antibiotics, which by themselves would not reach their intracellular targets [216,217]. There are two ongoing clinical trials designed to examine the efficacy of SPR741 against *P. aeruginosa* (**NCT03022175** and **NCT03376529**).

The first clinical trial (**NCT03022175**) is a phase I randomized control trial designed to evaluate the safety, tolerability, and pharmacokinetics of both single and multiple intravenous doses of SPR741 when given to 64 healthy adult volunteers. The trial consists of a single ascending dose (SAD) phase and a multiple ascending dose (MAD) phase. Participants in SAD will either receive a single dosage of SPR741 or a placebo. Over 14 days, participants in MAD will either receive repeated doses of SPR741 or a placebo. Sequential cohorts will experience escalating dosages of SPR741 in both phases. During the SAD phase, patients will receive single doses of SPR741 over 60 min intravenous infusion at different doses. In the MAD phase, patients will receive SPR741 over 60 min intravenous infusion three times a day [197]. The second clinical (**NCT03376529**) is a phase I randomized control trial designed to evaluate the drug–drug interaction, pharmacokinetics, safety, and tolerability of a single dose of SPR741 combined with each of three different antibiotics (ceftazidime or piperacillin/tazobactam or aztreonam) in 27 healthy volunteers [196]. Patients were administered either a single dose of SPR741 alone, a single dose of SPR741 in combination with one of three different partner antibiotics, or the partner antibiotic alone in a randomized sequence [196].

Another polymyxin B compound, SPR206 and SPR741, interacts with and increases the permeability of the *P. aeruginosa* outer membrane, thereby enhancing the accumulation of coadministered antibiotics [217,218]. The efficacy of SPR206 was examined in two randomized clinical trials (**NCT03376529** and **NCT03792308**). In the first study, SPR206 was examined in **NCT03792308** to assess the safety, tolerability, and pharmacokinetics of single and multiple intravenous doses of SPR206 when administered to 94 healthy adult volunteers [198]. During the SAD phase, patients received single doses of SPR206 via intravenous infusion for over 1 h. During the MAD phase, patients received SPR206 via IV infusion 1 h three times a day for 7 consecutive days or 1 h three times a day for 4 consecutive days. A second phase I clinical trial (**NCT03376529**) evaluated the drug–drug interaction, pharmacokinetics, safety, and tolerability of a single dose of SPR741 combined with each of three different partner antibiotics (ceftazidime or piperacillin/tazobactam or aztreonam) in healthy volunteers [196]. Both studies showed that upon coadministration, the pharmacokinetic profiles of SPR741 and the associated antibiotics remained unaltered [219]. In general, SPR741 was well tolerated. These findings support SPR741’s continued clinical development for the treatment of severe illnesses caused by *P. aeruginosa* bacteria [219].

Similar to the mechanism of SPR206, BRX-8 is designed to be less toxic to human cells and, thus, improve its application in patients [220]. A phase I randomized control trial (**NCT04649541**) was designed to assess the safety and tolerability of single and multiple intravenous doses of MRX-8. The study was focused on the pharmacokinetics of MRX-8 and its primary metabolite following single and multiple intravenous doses including the elimination rate of MRX-8 and its metabolite in urine [199].

#### 3.1.3. Beta-Lactamase Inhibitors

A brand-new beta-lactamase inhibitor, relebactam, targets upon several beta-lactamase enzymes in several multi-drug-resistant bacteria (Table 1B) [221,222]. Relebactam was added to imipenem in vitro to restore its effectiveness against various imipenem-resistant bacteria, including *P. aeruginosa*. In response, there have been several clinical trials (**NCT02493764, NCT02452047, NCT05561764, NCT03583333, and NCT05204563**) to examine the use of relebactam against *P. aeruginosa*. A phase III randomized control trial (**NCT02452047-RESTORE-IMI 1**) was designed to evaluate the efficacy and safety of imipenem + cilastatin/relebactam versus colistimethate sodium + imipenem + cilastatin in the treatment of imipenem-resistant *P. aeruginosa*. Infections evaluated in the study included hospital-acquired bacterial pneumonia, ventilator-associated bacterial pneumonia, complicated intra-abdominal infection, and complicated urinary tract infection [200,201]. Patients with hospital-acquired/ventilator-associated pneumonia, complicated intra-abdominal infection, or complicated urinary tract infection caused by imipenem-nonsusceptible (but colistin- and imipenem/relebactam-susceptible) pathogens received for 5–21 days imipenem/relebactam or colistin and imipenem [200,201]. In the clinical trial, 16 patients received colistin plus imipenem, while 31 patients received imipenem/relebactam. The study found that 70% of patients taking colistin and imipenem and 71% of patients on imipenem/relebactam showed a favorable overall response. The mortality rate at 28 days was 10% among the imipenem/relebactam-treated group and 30% in the colistin/imipenem-treated group. In addition, 10% of imipenem/relebactam patients and 31% of colistin/imipenem patients experienced serious adverse effects, with the most frequent being nephrotoxicity [200,201]. Overall, imipenem/relebactam was found to be both an efficacious and well-tolerated treatment option for carbapenem-nonsusceptible infections, particularly among *Pseudomonas* species. A subsequent phase III clinical trial (**NCT02493764-RESTORE-IMI 2 and NCT03583333**) compared treatment with a fixed-dose combination of imipenem/relebactam/cilastatin with a fixed-dose combination of piperacillin/tazobactam in participants with hospital-acquired or ventilator-associated bacterial pneumonia (HABP or VAPB, respectively) [202,204]. The overall goal was to assess whether imipenem/relebactam/cilastatin was noninferior to piperacillin/tazobactam [202,204]. Patients with HABP/VABP were randomized to receive piperacillin/tazobactam intravenously every 6 h for 7–14 days or imipenem/cilastatin/relebactam. All-cause mortality at day 28 served as the main outcome [202,204]. Of the 264 imipenem/cilastatin/relebactam and 267 piperacillin/tazobactam patients included in the study, there was a similar improvement in mortality, morbidity, and clinical symptoms among patients who received either drug regimen. Overall, the study showed that patients infected with Gram-negative bacterial pathogens, including *P. aeruginosa*, can be treated effectively with imipenem/cilastatin/relebactam, even in critically ill, high-risk patients [202,204].

Given the results of the **RESTORE-IMI 1 and 2** trials, another phase I clinical trial (**NCT05561764**) examined whether the combination of imipenem/cilastatin/relebactam was effective for *P. aeruginosa* infections in CF patients [205]. Specifically, the study assessed the pharmacokinetics and tolerability of imipenem/cilastatin/relebactam in 16 adolescent and adult patients with CF acute pulmonary exacerbations [205]. The CF patients received 10–14 days of imipenem/cilastatin/relebactam every 6 h with dose determined per renal function, with or without adjunctive aminoglycoside or fluoroquinolone therapy. The forced expiratory volume in 1 s (FEV1) was measured throughout the experiment to assess efficacy [205].

In addition, another novel beta-lactamase antibiotic, nacubactam, has shown promising results as an effective antibiotic against *P. aeruginosa* infections (**NCT02134834, NCT02972255, and NCT03182504**) [207,208,209]. In a single phase I clinical trial, the safety, tolerability, and pharmacokinetics of intravenous nacubactam were assessed using 40 healthy volunteers who received three different dose regiments. The study found that the use of nacubactam with other antibiotics, such as meropenem, did not affect its pharmacokinetics or side effects [207,208,209]. Overall, these findings support the continued clinical development of nacubactam and demonstrate the suitability of meropenem as a potential β-lactam partner for nacubactam.

### 3.2. Bacteriophages

The emergence of mutants that are resistant to most available antibiotics represents a serious global health crisis. As such, alternative approaches, such as phage therapy, have been extensively investigated. Phage therapy involves the purification of virulent lytic phages that infect and eliminate phage-sensitive bacteria [223]. Within the last several years, numerous clinical studies have been conducted to evaluate the effectiveness of phage therapy in treating specific *P. aeruginosa* infections.

A randomized, controlled, double-blind phase I/II trial was designed to test the efficacy of phage therapy for the treatment of *P. aeruginosa* wound infections in burned patients. The trial was given the name PhagoBurn (**NCT02116010**) (Table 2). In this study, a cocktail of 12 natural lytic *P. aeruginosa* bacteriophages (PP1131) was administered topically at very low concentrations and compared with the standard of care (1% sulfadiazine silver emulsion cream) [224]. Eligible participants were randomly assigned (1:1) to receive either the standard of care (1% sulfadiazine silver emulsion cream) or a cocktail of 12 natural lytic anti-*P. aeruginosa* bacteriophages (PP1131 at a concentration of 1 × 10^6^ plaque-forming units (PFU) per mL) [224]. Both treatments were administered topically daily for 7 days with a 14-day follow-up period [224]. The main aim of the clinical trial was to determine the median time to a sustain and decrease in bacterial burden by at least two quadrants using a four-quadrant method [224]. This was performed using daily swabs on all participants who had a microbiologically confirmed infection on day 0 and who received at least one sulfadiazine silver or phage dressing. In total, 27 patients were enrolled across two recruiting periods totaling 13 months; 13 patients were randomly assigned to a phage therapy and 14 to the standard of care [224].

According to the PhagoBurn clinical trial and by the end of the phage treatment, half of the subjects successfully reduced their daily bacterial burden in most infected lesions by two quadrants or more [224]. However, the median time to reach this endpoint for those who received PP1131 was considerably longer than it was for those who received the standard of care [224]. The systemic administration of antibiotics that are effective against the infecting strain, whether they were started at day 0 or added later during the study therapy, had no impact on this outcome [224]. Investigators suggested that even though the phage multiplied on *P. aeruginosa* phage-sensitive strains within the infected burn wound, the phage cocktail titer was very low. In addition, due to challenges in manufacturing the cocktail, the patient sample size was small. The authors of the study suggested that additional studies using increasing phage concentrations and larger sample sizes may improve the outcomes for burn patients.

The study “MUCOPHAGES” (**NCT01818206**) examined the impact of a cocktail of 10 phages on *P. aeruginosa* strains isolated from CF patients’ sputum samples [225]. The trial’s aim was to evaluate the efficacy of bacteriophages in eliminating infecting *P. aeruginosa* strains present in sputum samples from CF patients who were 6 years and older [225]. Subsequently, a cocktail of 10 bacteriophages was applied directly to 60 sputum samples obtained from CF patients. The investigators determined that the number of *P. aeruginosa* (CFU) within each sample after 6 h alters the applications of the phages [225].

Another clinical trial is being conducted to assess the efficacy of a phage cocktail B-PAO1, which consists of four phages, against life-threatening *P. aeruginosa* infections (**NCT03395743**) [226]. The main aim of this clinical trial was to allow physicians to provide treatment with investigational drug, AB-PA01, for patients with serious or immediately life-threatening *P. aeruginosa* infections, for which no alternative treatments are currently available [226]. However, further studies with larger populations are still required to prove the efficacy of using bacteriophage therapy in treating *P. aeruginosa* infections. Based on the clinical trial data, investigators recommended modifications of both the composition and the application of the phage cocktail to improve therapeutic outcomes.

Another phase I clinical trial evaluated the safety and tolerability of a phage cocktail-SPK therapy compared with the standard of care (Xeroform and Kenacomb) for second-degree burn wounds infected by *S. aureus*, *P. aeruginosa*, or *K. pneumoniae* in adult patients (**NCT04323475**) [227]. The study included 12 adult patients who were split into control and treatment groups. The control group consisted of Xeroform primary dressing and Kenacomb topical antibiotic cream (for wounds with signs of localized infection) [227]. The experimental group consisted of dosage-metered airless spray containing a cocktail at a concentration of 1.4 × 10^8^ PFU/mL for an effective dosage of 2.5 × 10^5^ PFU/cm^2 of burned area [227].

A phase I randomized clinical trial (**NCT04596319**) was designed to evaluate the safety, tolerability, and phage recovery profile of inhaling the AP-PA02 multi-bacteriophage to treat 29 CF patients and/or chronic pulmonary *P. aeruginosa* infections for a period of 4 weeks [228]. Another phase II clinical trial (**NCT04684641**) examined whether YPT-01 phage therapy reduces sputum bacterial load in 8 CF subjects with *P. aeruginosa* for 7 days. In addition, the study evaluates the safety profile of phage therapy in CF patients [229].

### 3.3. Strategies Targeting P. aeruginosa Virulence (Biofilm, Quorum Sensing, Type III Secretion System, and Antimicrobial Peptides)

#### 3.3.1. Quorum Sensing

The role of QS in *P. aeruginosa* virulence has been well established in animal models [230,231,232,233]. A clinical trial testing the ability of inhaled azithromycin to inhibit *P. aeruginosa* QS began in 2005 (Table 3). Previous preclinical studies showed that azithromycin inhibits the transcription of several genes involved in the function of QS in *P. aeruginosa* [234,235,236,237]. Specifically, azithromycin inhibits the 23S rRNA of the 50S ribosome unit, which decreased the production of genes essential for QS functions in *P. aeruginosa* [234,235,236,237]. The study was a prospective analysis of 92 intubated patients who were colonized with *P. aeruginosa* in intensive care units at 17 European hospitals [238]. Throughout the study, tracheal isolates were collected each day to estimate the total density of *P. aeruginosa* bacteria in the aspirates through genomic copy numbers [238]. The study showed that azithromycin, which has no bactericidal activity against *P. aeruginosa* but interferes with its QS, reduced QS-gene expression in tracheal aspirates of treated patients (**NCT00610623**) [238]. The study also showed that the prevalence of noncooperating (and hence less virulent) lasR *P. aeruginosa* mutants rose with time in the absence of azithromycin [238]. The LasR protein is a transcriptional activator that regulates the expression of several of the *P. aeruginosa* QS-controlled genes, including lasB, lasA, las, and phenazines [238]. In tracheal aspirates, direct QS-gene expression was considerably lowered by azithromycin. During azithromycin treatment, the benefit of lasR-mutants was lost, and virulent wild-type isolates predominated [238]. These in vitro findings were supported with the observation that the growth of the wild-type strain and not the LasR mutants decreased by azithromycin [238]. Furthermore, the absence of azithromycin prevented the lasR-mutant from successfully encroaching on wild-type populations [238]. The authors suggested that intervention based on QS blockade may increase the prevalence of *P. aeruginosa* strains with more virulent genotypes within the hospital environment [238]. A subsequent study by Welsh et al. corroborated these findings and suggested that alterations in virulence phenotype occur when only one compound is used to block a specific component of QS [239]. Despite these problems, it is likely that attempts to target QS to reduce *P. aeruginosa* virulence will continue.

#### 3.3.2. Antibiofilm Agent

The antibiofilm agent OligoG has shown promise as an effective agent against *P. aeruginosa* infection [244,245]. By chelating calcium, the alginate oligosaccharide OligoG has the potential to reduce the viscosity of CF patients’ sputum, facilitate mucus removal from patient airways, and lower microbial burden and inflammation [244,245]. OligoG disrupted the biofilm structure of the *P. aeruginosa* mucoid phenotype, which may enhance the function of the host immune system and the efficacy of antibiotics. A phase I trial was conducted with a focus on pulmonary function and adverse events to investigate the safety and local tolerability of multiple dosages of OligoG administration of an inhaled fragment in 26 healthy volunteers (**NCT00970346**) [240,244,245]. Another phase II/III clinical trial (**NCT03822455 and NCT03698448**) assessed the safety, efficacy, and tolerability of OligoG in CF patients for 12 weeks [241,246].

#### 3.3.3. Antimicrobial Peptides

Antimicrobial peptides are bioactive compounds that are extremely biocompatible and comparatively resistant to the development of bacterial resistance [247,248,249,250]. Most antimicrobial peptides eliminate bacteria by altering the normal permeability of the cell membrane [247,248,249,250]. In addition, there are several antimicrobial peptides that have been effectively covalently immobilized on a range of surfaces, including silicone, glass, titanium oxide, resin beads, and contact lenses [247,248,249,250]. In addition, PLG0206 (WLBU2), a broad-spectrum engineered cationic antimicrobial peptide with broad-spectrum activity, inhibits in vitro *P. aeruginosa* biofilm growth on airway epithelial cells [242]. PLG0206 is an engineered antibacterial peptide that is based on naturally occurring antimicrobial peptide and has shown effectiveness against *P. aeruginosa* and *staphylococcus aureus* biofilms [242]. A phase I trial was conducted in 14 healthy volunteers in 2018. In 2022, PLG0206 was entered into a phase I clinical trial to treat *P. aeruginosa* infections of the prosthetic joints (**NCT05137314**) [242].

#### 3.3.4. Type III Secretion System

Numerous studies using both the murine models and the ex vivo model demonstrated the role of T3SS in the virulence of *P. aeruginosa* [127,251,252,253,254,255,256]. Several inhibitors of T3SS, including small molecule proteins and carbohydrates, have been discovered [257,258,259]. In general, these inhibitors target the regulation of different genes associated with essential functions of T3SS [257,258,259]. Fluorothiazinon is a small molecule that targets the function of T3SS [257,258,259]. Fluorothiazinon belongs to a class of 2,4-disubstituted-4H-[1,3,4]-thiadiazine-5-one molecules that inhibited the function of T3SS in chlamydia and salmonella [257]. Further analysis revealed that Fluorothiazinon inhibits ExoT and ExoY secretion in *P. aeruginosa* [257,258]. In addition, the molecule reduces bacterial cytotoxicity and enhances bactericidal internalization by both epithelioid and phagocytic cells [257,258]. Furthermore, Sheremet et al. demonstrated that Fluorothiazinon decreased bacterial load in the lung by eliminating *P. aeruginosa* and decreasing levels of IL-6, TNF **α,** and interferon **γ** [257,258]. In 2018, the Gamaleya Research Institute (Ministry of Health of the Russian Federation, Rakhmanovsky pereulok 3, Tverskoy District, Moscow) initiated a randomized placebo-controlled phase II trial consisting of 777 patients to evaluate the safety and efficacy of the drug Ftortiazinon (aka Fluorothiazinon) with cefepime in comparison with placebo and cefepime in the treatment of hospitalized adult patients with complicated urinary tract infections caused by *P. aeruginosa* (**NCT03638830**) [243]. The 777 patients were randomized into three separate groups: 150 mg Ftortiazinon, placebo, and cefepime; 300 mg Ftortiazinon and cefepime; and placebo and cefepime [243].

### 3.4. P. aeruginosa Virulence Factor Passive Immunotherapy

Originally, opsonic antibodies against the mucoid exopolysaccharide (MEP) of *P. aeruginosa* were shown to be protective in a murine model of chronic respiratory infection (Table 4) [260].

After opsonic *P. aeruginosa* MEP antibodies were successfully produced in mice [275], the immunogen was used to elicit opsonizing anti-MEP antibodies in rats [276]. These harvested antibodies, MEP IVIG, were used in a phase II clinical trial aimed at reducing acute pulmonary exacerbation of *P. aeruginosa* infection in CF patients. The trial was concluded in 2000, but no results ensued (**NCT00004747**) [261]. The clinical trial studied the efficacy of monthly intravenous mucoid exopolysaccharide *P. aeruginosa* immune globulin (MEP IGIV) given over 1 year in reducing the frequency of acute pulmonary exacerbation in patients with cystic fibrosis, mild to moderate pulmonary disease, and mucoid *P. aeruginosa* colonization [261]. The study examined whether MEP IGIV improved: forced expiratory volume in the first second (FEV1), reduced the density of mucoid *P. aeruginosa* colonies in the patient’s sputum, and improved the quality of life in patients infected with *P. aeruginosa* [261]. A total of 170 patients were included in the study to be randomized into three treatment groups: low-dose intravenous mucoid exopolysaccharide *P. aeruginosa* immune globulin (MEP IVIG), high-dose MEP IVIG, or placebo [261]. The treatments were administered every 28 days for a year [261].

Immunotherapeutic approaches targeting T3SS of *P. aeruginosa* have also been pursued. Passive immunization using PcrV, a protein critical for the translocation of T3SS effectors, significantly enhanced the survival of mice in lung and burn models of *P. aeruginosa* infection [277,278]. Subsequently, anti-PcrV MAb F(Ab′)2 monoclonal antibodies were generated and shown to be protective in murine models of lung infection [279,280]. Anti-PcrV MAb 166 has been developed for clinical use as KB001, an anti-PcrV PEGylated MAb F(Ab′)2 (**NCT00638365, NCT00691587, and NCT01695343**) [262,263,264,279]. A clinical trial (**NCT00691587**) evaluated KB001 in patients in the intensive care setting who were receiving ventilator therapy and suffering from *P. aeruginosa* lung infections. Patients received either the placebo or one of two dose levels of KB001. KB001 (Humaneered™), which was used to treat *P. aeruginosa* infections, is a modified, PEGylated, recombinant anti-*P. aeruginosa* PcrV Fab’ antibody that binds with high affinity to the PcrV protein and blocks its activity [262,263]. The transport of *P. aeruginosa* exotoxins into host immunological and epithelial cells depends on the protein PcrV, which is located close to the tip of the TTSS needle [263]. As such, this represents a novel therapeutic strategy for treating infection by reducing inflammation in CF patients through the inhibition of the PcrV protein’s function, thereby interfering the release of powerful cytotoxins involved in the initiation and maintenance of *P. aeruginosa* infections [263]. The clinical trial examined whether KB001 protects host epithelium and immune cells, and evaluated the reduction of pulmonary *P. aeruginosa* burden [262]. Patients were randomized to receive a low dose of KB001 monoclonal antibody, high dose of KB001 monoclonal antibody, or placebo [262].

A phase I/II clinical trial studied the safety, pharmacokinetic, and pharmacodynamic properties of KB001 in CF patients with chronic *P. aeruginosa* infections (**NCT00638365**) [263]. A single intravenous dosage of either KB001 (3 or 10 mg/kg) or a placebo was administered to 27 participants of at least 12 years of age [263]. In each patient, assessments were made concerning KB001’s safety and pharmacokinetics. The assessment also included an examination of the *P. aeruginosa* density, clinical results, and inflammatory markers [263]. KB001 had an acceptable safety profile. After a single dose, there were no appreciable differences between KB001 treated and the placebo groups in either: *P. aeruginosa* density, symptoms, or spirometry [263]. At day 28, sputum myeloperoxidase, IL-1, and IL-8, sputum neutrophile elastase, and neutrophil counts showed a dose-dependent reduction in the KB001 (10 mg/kg)-treated CF patients compared with the placebo group [263]. The clinical trial showed that KB001 targeted *P. aeruginosa* TTSS in CF patients with persistent *P. aeruginosa* infection; it also decreased both airway inflammation and lung damage.

A phase I/II clinical trial also examined whether 16 weeks of KB001 treatment improved the time required for the use of antibiotics in patients with worsening respiratory symptoms. In addition, the trial examined KB001’s safety and efficacy in improving inflammatory markers and spirometry for patients with a history of chronic *P. aeruginosa* infections (**NCT01695343**) [264]. At least one infusion of KB001 (*n* = 83) or a placebo (*n* = 86) was given to a total of 169 patients [264]. Except for one serious adverse event involving increased liver enzymes, KB001 was generally safe and well tolerated when compared with placebo [264]. The time before an antibiotic was required was the same for all groups. At week 16, KB001-A outperformed the placebo, resulting in an increase in percent predicted forced expiratory volume in 1 s [264]. At week 16, sputum neutrophil elastase showed a nonsignificant decline, whereas IL-8 concentrations were considerably lower in KB001-treated patients than in the placebo group [264].

The Psl exopolysaccharide is another virulence factor present in numerous strains of *P. aeruginosa* [281]. Antibodies against Psl were detected by phenotypic screening of human antibody phage display libraries constructed from peripheral blood B cells of healthy volunteers and individuals recovering from *P. aeruginosa* infections [282]. The Psl monoclonal antibody MAb Cam-003 was further evaluated in the acute lethal pneumonia murine model in which the antibody provided significant protection against *P. aeruginosa* strains expressing PsL [282]. It was later postulated that anti-PcrV would benefit by combination with another antivirulence factor antibody [283]. A bispecific antibody, BiS4αPa, targeting both PcrV and Psl was developed and shown to be protective in the murine pneumonia model, which led to the development of the clinical candidate MEDI3902 (**NCT02255760 and NCT02696902**) [265,266,283]. In a phase I trial (**NCT02255760**), healthy patients between the ages of 18 and 60 were given a single intravenous infusion of MEDI3902 to assess the drug’s safety, pharmacokinetics, antidrug antibody responses, ex vivo anticytotoxicity, and opsonophagocytic killing capabilities against *P. aeruginosa*. Using randomization, 56 patients were randomly assigned to receive either 250, 750, 1500, or 3000 mg of MEDI3902 or a placebo over a period of 60 days [265]. There were no major adverse events. The pharmacokinetics of MEDI3902 were roughly linear for doses of 250, 750, and 1500 mg and nonlinear for dosages of 1500 and 3000 mg. All doses of MEDI3902 were linked with serum anticytotoxicity antibody concentrations and opsonophagocytic killing activity [265]. The results of MEDI3902’s phase I study in healthy volunteers justify further investigation of the drug’s efficacy and safety in patients at risk for *P. aeruginosa* pneumonia.

A phase II clinical trial examined the efficacy, pharmacokinetics, and safety of MEDI3902 in mechanically ventilated ICU patients infected with *P. aeruginosa* (**NCT02696902**) [266]. The study included 168 patients with PCR-confirmed *P. aeruginosa* colonization of the lower respiratory tract who were randomized into either a single IV infusion of 1500 mg MEDI3902 (*n* = 85) or placebo (*n* = 83) [266]. The study showed that a single IV dosage of MEDI3902 increased the drug concentration over the desired amount, but it did not reduce *P. aeruginosa* pneumonia as the primary efficacy outcome.

In another study, eggs from chickens that have received a *P. aeruginosa* vaccination were used to create “anti-pseudomonas IgY”, called PseudIgY (avian Ab to *P. aeruginosa*). The trial assessed whether daily gargling with PsAer-IgY against *P. aeruginosa* prevents *P. aeruginosa* infections in CF patients (**NCT00633191**) [267]. The study included 14 CF patients who experienced sporadic *P. aeruginosa* infections. In the trial, patients received a brief course of antibiotics to eliminate *P. aeruginosa* infection. After that, they began gargling with a PseudIgY solution every night to ward off infection [267].

Another phase III double-blind placebo-controlled trial investigated whether PseudIgY prolongs the time to reinfection with *P. aeruginosa* following a successful treatment of acute or intermittent infection (**NCT01455675**) [268]. The PseudIgY and the control groups were told to gargle and ingest either solution [268]. In the treatment group, the patients were required to gargle for 2 min 70 cc of the IgY/placebo solution every night for 2 years. The study enrolled 164 patients to be randomized into either group [268].

Several immunotherapeutic trials have been conducted targeting different antigens of *P. aeruginosa*. One of the targeted antigens was the O-polysaccharide moiety of *P. aeruginosa* serotype O11. A human monoclonal IgM antibody, KBPA-101 or panobacumab, was developed and found to be safe and efficacious in preclinical in vitro and in vivo studies and in phase I clinical trials (**NCT00851435**) [269,270,271]. In the trial, 14 patients who did not receive the antibody were compared with 17 panobacumab patients (13 of whom received the complete course of treatment, consisting of three doses of 1.2 mg/kg). Within the group that received the complete three-course panobacumab treatment, adjunctive immunotherapy led to improved clinical outcomes [269,270,271]. Specifically, patients that received panobacumab had a greater resolution rate of 85% (11/13) compared with 64% (9/14) in the place group. Compared with the control, patients who received panobacumab also had a shorter time to clinical resolution [269,270,271]. In addition, panobacumab was safe and resulted in a high clinical cure and survival rates in patients developing nosocomial *P. aeruginosa* O11 pneumonia [269,270,271].

More recently, a phase II clinical trial was conducted to assess the efficacy, safety, and pharmacokinetics of a new human monoclonal antibody against *P. aeruginosa* alginate, AR-105 (Aerucin^®^), as adjunctive therapeutic treatment to standard-of-care antibiotics for *P. aeruginosa* pneumonia in mechanically ventilated patients (**NCT03027609**) [272]. The study was conducted at approximately 100 clinical sites across 17 countries [272]. The 158 patients included in the clinical trial were split into the control group or those given one intravenous infusion of AR-105 20 mg/kg.

Recent studies were conducted to elucidate the mechanism of action of IgM-enriched immunoglobulin (IgM-IVIg) [284]. The studies involved utilizing LPS from *E. coli* to cause sepsis in two animal models: Syrian golden hamsters [285] or the CLP model to induce sepsis in rats [286]. Further studies in humans have been conducted to elucidate the mechanism of action of IgM-IVIg [287]. All three studies concluded that the administration of IgM-IVIg (Pentaglobin^®^) attenuated the endotoxin (LPS) activity by reducing inflammation. A clinical trial examined the efficacy of IgM-enriched intravenous immunoglobulin (Pentaglobin^®^—5 mL/kg over a 12 h intravenous infusion for 3 consecutive days) to decrease mortality in 120 neutropenic acute leukemia or hematopoietic stem cell transplant patients colonized with carbapenem-resistant *Enterobacteriaceae* or *P. aeruginosa* (**NCT03494959**) [273].

Another promising novel treatment, TRL1068, a human monoclonal antibody that targets DNA-binding protein II (DNABII), is currently undergoing clinical testing. DNABII is an important structural element of biofilms because it stabilizes extracellular DNA [288,289,290]. This monoclonal antibody is being investigated for use in prosthetic joint infections, particularly with multi-drug-resistant *P. aeruginosa* species [288,291].Though rare, prosthetic joint infections caused by Gram-negative bacilli, such as *P. aeruginosa*, is a catastrophic consequence that leads to prolonged hospitalization and higher medical costs [292,293,294]. Specifically, *P. aeruginosa* is linked to osteomyelitis, septic arthritis, and prosthetic joint infections due to its ability to adhere to bone and fibrocartilaginous articular structures. A phase I randomized clinical trial (**NCT04763759**) was designed to assess the overall safety and pharmacokinetics of TRL1068 in 18 patients with prosthetic joints using three separate dosages: 6, 15, and 30 mg/kg [274]. The study will measure the number of *P. aeruginosa* and the level of C-reactive protein (CRP), erythrocyte sedimentation rate (ESR), IL-6, and IL-10 [274].

### 3.5. P. aeruginosa Outer Membrane Proteins as a Vaccine

The first trial (NCT00778388) involved active immunization with a *P. aeruginosa* hybrid protein, OprF/I (Table 5). In the hybrid protein, the *P. aeruginosa* outer membrane OprF is fused with the lipoprotein OprI. Active immunization of neutropenic mice with OprF/I was protective against lethal *P. aeruginosa* [295,296]. In addition, the anti-OprF/I antibody protected mice with severe combined immunodeficiency from *P. aeruginosa* infection. Furthermore, OprF/I affected the *P. aeruginosa* QS system and the QS-related virulence in the plant and worm models [295,296].

A phase I trial examined the immunogenicity, safety, and tolerance in a healthy adult immunized with three different dosages of IC43 compared with placebo (**NCT00778388**) [297]. Two intramuscular injections of IC43 were administered in the deltoid region 7 days apart, and the patients were randomly assigned to one of five treatment groups: 50, 100, or 200 g IC43 with adjuvant, 100 g IC43 without adjuvant, or placebo (0.9% sodium chloride) [297]. In healthy volunteers, IC43 dosages of 50 g or more were sufficient to cause a plateau in IgG antibody responses [297].

A subsequent phase II study in which IC43 was used to immunize mechanically ventilated ICU patients has been completed using doses of 100 and 200 μg with adjuvant and a dose of 100 μg without adjuvant (**NCT00876252**) [298]. Patients were initially randomized to IC43 100 μg, IC43 200 μg, or placebo. On day 0, patients were immunized. On day 7, a second immunization was administered. Up until day 90, clinical study visits were conducted. The OprF/I specific immunoglobulin IgG antibody titer was the primary objective in the immunogenicity evaluation at day 14 [298]. At visits made in the ICU, surveillance cultures were obtained for the identification of *P. aeruginosa* from the blood, wounds, respiratory tract, urine, and central venous catheter [298]. When medically necessary, the investigator collected samples for *P. aeruginosa* diagnosis between each visit up to day 90 [298]. All IC43 groups had higher OprF/I IgG antibody titers on day 14 compared with placebo. With 100 μg IC43 alone, seroconversion was greatest (80.6%) [298]. *Pseudomonas aeruginosa* infection rates were not significantly different, and the IC43 groups had a low incidence of invasive infections (11.2%–14.0%; pneumonia) [298]. In the group of patients receiving 100 μg of IC43 with adjuvant, 2 patients (1.9%) reported experiencing serious adverse effects thought to be connected to the treatment [298]. No fatalities were connected to trial therapy, and both serious adverse events resolved [298]. Overall, IC43 produced a significant immunogenic effect; however, *P. aeruginosa* infection rates did not differ significantly among the four treatment groups [298].

Another double-blind phase II/III clinical trial examined the efficacy, immunogenicity, and safety of IC43 recombinant *P. aeruginosa* vaccine in nonsurgical ICU patients (**NCT01563263**) [299]. Eight hundred patients between the ages of 18 and 80 who were admitted to the ICU and anticipated needing mechanical ventilation for less than 48 h were randomized 1:1 to receive two doses of IC43 100 g or a saline placebo, spaced 7 days apart [299]. All-cause death in individuals 28 days following the initial vaccination served as the primary efficacy outcome. Safety and immunogenicity were also assessed [299]. At day 28, the all-cause mortality rates in the IC43 and placebo groups were 29.2% and 27.7%, respectively [299]. Both groups’ overall survival rates and the percentage of patients with less than one confirmed *P. aeruginosa* invasive infection or respiratory tract infection were similar [299]. Compared with the IC43 100 μg vaccinated group (93.1%), more patients in the placebo group (96.5%) experienced one adverse event [299]. Respiratory failure (6.9% vs. 5.7%, respectively), septic shock (4.1% vs. 6.5%), cardiac arrest (4.3% vs. 5.7%), multiorgan failure (4.6% vs. 5.5%), and sepsis (4.6% vs. 4.2%) were the most frequently reported experienced serious adverse events in the IC43 and placebo groups [299]. In the IC43 treated group, there were no linked significant adverse events recorded [299]. The IC43 100 g vaccination was well tolerated [299]. High immunogenicity was achieved by the vaccine; however, there was no therapeutic advantage above placebo in terms of reducing total mortality [299].

### 3.6. Strategies Targeting P. aeruginosa Iron Acquisition Systems

In addition to the previous treatments, other alternatives for eliminating *P. aeruginosa* infections have been investigated, including inhaled gallium and sodium nitrite (Table 6) [157,300,301]. A previous preclinical study showed that gallium works by decreasing bacterial iron (Fe) uptake and interfering with Fe signaling [302]. A phase I clinical trial examined the pharmacokinetics, safety, and tolerability of an intravenous infusion of a drug called Ganite^®^ (IV gallium nitrate) in *P. aeruginosa* infected CF (**NCT01093521**) [157]. The trial included CF patients with chronic *P. aeruginosa* respiratory infections [157]. The 20 patients involved in the study were randomized into two groups that received continuous IV gallium nitrate (Ganite^®^) infusion for 5 days with either 100 or 200 mg/m^2^.

A subsequent phase II, multicenter, randomized, placebo-controlled trial in adult CF patients chronically infected with *P. aeruginosa* was conducted to evaluate the efficacy of IV gallium in improving pulmonary function as measured by a 5% or greater relative improvement in forced expiratory volume in 1 s (FEV1) from baseline to day 28 (**NCT02354859**) [300]. The 119 patients were randomized into two groups: continuous infusion of either gallium nitrate (200 mg/m^2^/day) or normal saline over 5 days [300].

In addition to gallium, another phase I/II study assessed the safety of inhaled sodium nitrite in 35 adults with CF and chronic *Pseudomonas* infections to reduce the burden of *P. aeruginosa* **NCT02694393**) [301]. Sodium nitrite was effective in eliminating *P. aeruginosa* within the lung of CF-infected patients by increasing the susceptibility of *P. aeruginosa* to antibiotics [304,305]. Patients inhaled 46 or 80 mg of sodium nitrite twice daily for 4 weeks.

Cefedrolor is a novel siderophore cephalosporin with broad activity and high potency against multi-drug-resistant Gram-negative bacteria [303,306]. A phase III randomized clinical trial (**NCT03032380**) was designed to compare all-cause mortality at day 14 in patients receiving cefiderocol or meropenem for HABP, VABP, or health-care-associated bacterial pneumonia (HCABP) caused by Gram-negative pathogens [303,306]. Between 7 to 14 days, patients received 3 h intravenous infusions of either cefiderocol (*n* = 148) or meropenem (*n* = 152) every 8 h. Moreover, open-label intravenous linezolid was administered to all patients for a minimum of 5 days [303,306]. In patients with Gram-negative nosocomial pneumonia, cefiderocol had similar tolerability and was noninferior to high-dose, extended-infusion meropenem in terms of all-cause mortality on day 14. The findings suggest that cefiderocol is a feasible therapy choice for individuals with nosocomial pneumonia, even those infected with multi-drug-resistant Gram-negative bacteria [303,306].

## 4. Conclusions

It is evident from the outcomes of the different clinical trials described in this review that the road to developing an effective anti-*P. aeruginosa* vaccine or therapy is long and challenging. Antibiotics will still be essential in combating *P. aeruginosa* infections. For example, other clinical trials have examined newer combinations of beta-lactamases, polymyxins, and alternative formulations, such as liposomal or inhalation, for traditional antibiotics against *P. aeruginosa* infections [201,306,307,308,309,310,311,312,313,314,315,316,317,318,319,320,321]. However, the continuous use of these antibiotics will likely lead to the emergence of multi-drug-resistant *P. aeruginosa* mutants. Therefore, developing and testing several nontraditional therapies is critical. These therapies may not be alternatives to or replace the current antibiotics. Rather, they may be used in conjunction with antibiotics to help reduce the dose of antibiotics required to treat infected patients, which will lessen the production of *P. aeruginosa* bioburden and/or reduce the rate of emergence of antibiotic resistant mutants.

Besides antibiotics, anti-*P. aeruginosa* treatments described in this review include: bacteriophage; vaccines targeting outer membrane proteins (opRF/opRI); and treatment targeting an essential virulence factor, such as the QS system, T3SS, and monoclonal antibodies. As we show in this review, many of these approaches produced encouraging results. In addition, and except for bacteriophages, these treatments are less likely to exert pressure on *P. aeruginosa* to develop resistant mutants. Bacteriophage therapy shares with antibiotics the inherit problem of emerging phage- or antibiotic-resistant mutants that may result from the overuse of these bacteriophages. However, unlike antibiotics, the application of multiple phages (cocktails) will significantly reduce the problem associated with the emergence of antibiotic resistant mutants.

Among the different nontraditional therapies described in this review, the most promising one is targeting a specific *P. aeruginosa* virulence factor, such as outer membrane proteins or a system/mechanism that could help the production of multiple virulence factors, such as the QS system, T3SS, and iron-acquisition system. However, a key factor that needs to be considered in developing these therapies is the pathogenesis of *P. aeruginosa* in different acute and chronic infections, including acute pneumonia, VAP, burn wound infections, urinary tract infections, and chronic lung infections in CF patients. Several previous studies have already demonstrated that the capacity of *P. aeruginosa* to cause different infections lies in its ability to tailor the production of different cell-associated and extracellular virulence factors in response to environmental cues within certain infection sites, including lung wounds, blood, or urinary tract. Therefore, to further develop current therapies and to identify new targets for further therapies, it is critical to define a specific virulent factor(s) that is produced in response to the in vivo environment. Previous studies analyzed *P. aeruginosa* pathogenesis utilizing growth conditions that mimic the in vivo conditions. For example, Palmer et al. [322] utilized CF sputum to analyze the pathogenesis of *P. aeruginosa* during chronic lung infections of CF patients, while Kruczek utilized whole blood to examine the pathogenesis of *P. aeruginosa* infections in severely burned patients [253]. Palmer et al. showed that compared with a laboratory medium, the growth of *P. aeruginosa* in CF sputum differentially expressed the iron-acquisition genes and flagella motility genes and enhanced the production of the cell-to-cell communication molecule, PQS [322]. In addition, CF sputum enhanced the production of the cell-to-cell communication molecule PQS [322]. As we showed in this review, gallium-related therapy that was assessed in clinical trials targeted the *P. aeruginosa* iron-acquisition system. Additional anti-*P. aeruginosa* therapies to treat lung infections in CF patients may target the synthesis and/or the secretion of the PQS molecule. Similarly, Kruczek et al. showed that compared with the growth in blood from healthy volunteers, the growth of *P. aeruginosa* in blood from severely burned patients significantly increased the expression of more than 1000 genes, including genes encoding the QS-controlled virulence factors and those encoding the transport heme and phosphate. Any of these systems is a likely target for further therapy to prevent bacteremia and sepsis in *P. aeruginosa*–infected severely burned patients.

As we demonstrated in this review, several clinical trials were focused on assessing the effectiveness of new antibiotics targeting the PcrV protein (Table 4). The 32.3 kDa PcrV is one of the several proteins that constitute T3SS in *P. aeruginosa*. The PcrV protein plays a pivotal role in the pathogenesis of *P. aeruginosa* infections. The production of several cell-associated and extracellular virulence factors strongly indicates the reliance of *P. aeruginosa* on many of these factors in establishing infection at different sites within the host. However, despite these factors, previous studies using the murine model of lung infection and the murine model of thermal injury clearly showed that specific PcrV antibodies only compromised the ability of *P. aeruginosa* to cause lung infection and bacteremia/sepsis, respectively [277,323]. Based on the result of these studies, more monoclonal antibiotics were developed, and their effectiveness in targeting *P. aeruginosa* lung infections in CF patients was assessed using the clinical trials described in Table 4. However, so far, the results of these trials are not encouraging (Table 4). In the murine model of thermal injury, PcrV antibodies eliminated bacteremia and sepsis in the thermally injured mice but did not significantly reduce the *P. aeruginosa* bacterial load at the infection site [277]. This suggests that in severely burned patients, PcrV antibodies interfere with the *P. aeruginosa*–induced bacteremia/sepsis by affecting the translocation of *P. aeruginosa* into the bloodstream. Therefore, it is possible that future clinical studies may prove that the PcrV monoclonal antibodies are more effective in preventing *P. aeruginosa* bacteremia in severely burned patients than effectively treating *P. aeruginosa* lung infections in CF patients. In addition, future studies may identify a potential target, similar to PcrV, that plays a critical role in the pathogenesis of *P. aeruginosa* infections. A treatment containing a combination of antibodies to PcrV and such a target may prove to be more effective than PcrV antibody alone in treating lung infections in CF patients. The antibodies may also be utilized in synergy with currently available antibiotics in treating *P. aeruginosa* lung infections.

## Figures and Tables

**Figure 1 microorganisms-11-00916-f001:**
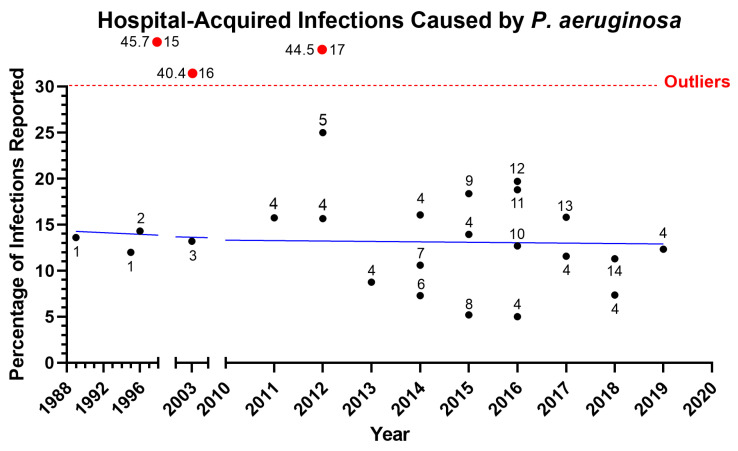
Infection rates of *P. aeruginosa* in hospitalized patients over the past 3 decades (1989–2019). Hospital-acquired infections reported included bloodstream infections and catheter-associated bloodstream infections, catheter-associated urinary tract infection, surgical site infections and burn wound infections, and hospital-associated and ventilator-associated pneumonia; although not every article reported every category, all except two included three or more categories. The blue line is a regression line for the data excluding outliers, which are shown above the dotted red line. The numbers associated with the black circles indicate reference numbers; the numbers to the left of the red circles are the percentages, and those to the left are the reference numbers. **References:** 1. [5], 2. [6], 3. [7], 4. [8], 5. [9], 6. [10], 7. [11], 8. [12], 9. [13], 10. [14], 11. [15], 12. [16], 13. [17], 14. [18]. Outliers: 15. [19], 16. [20], 17. [21].

**Figure 2 microorganisms-11-00916-f002:**
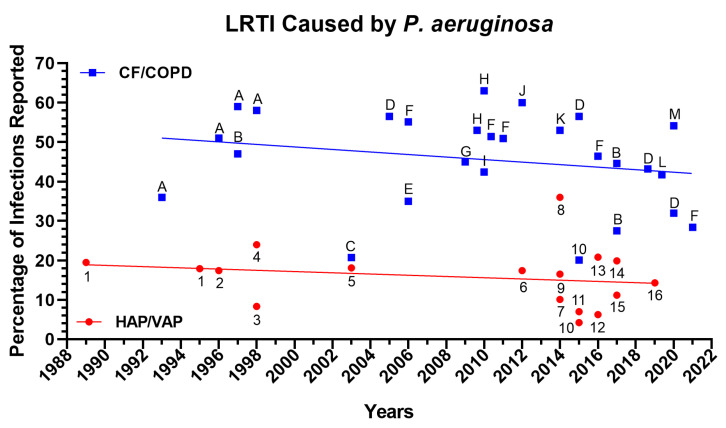
Lower respiratory tract infection (LTRI) rates of *P. aeruginosa*. Reported rates of hospital-associated and ventilator-associated pneumonia (HAP/VAP) over the past 3 decades (1989–2019) and the regression line for the data are in red. The rates for patients with cystic fibrosis (CF) or chronic obstructive pulmonary disease (COPD) and the associated regression line are in blue. The numbers and letters associated with the red circles and blue squares, respectively, indicate the reference numbers. **References for HAP/VAP:** 1. [5], 2. [6], 3. [22], 4. [19], 5. [7], 6. [21], 7. [11], 8. [23], 9. [10], 10. [24], 11. [12], 12. [15], 13. [16], 14. [17], 15. [25], 16. [8]. **References for CF/COPD**: A. [26], B. [27], C. [28], D. [29], E. [30], F. [31], G. [32], H. [33], I. [34], J. [35], K. [36], L. [37], M. [38], 10. [24].

**Figure 3 microorganisms-11-00916-f003:**
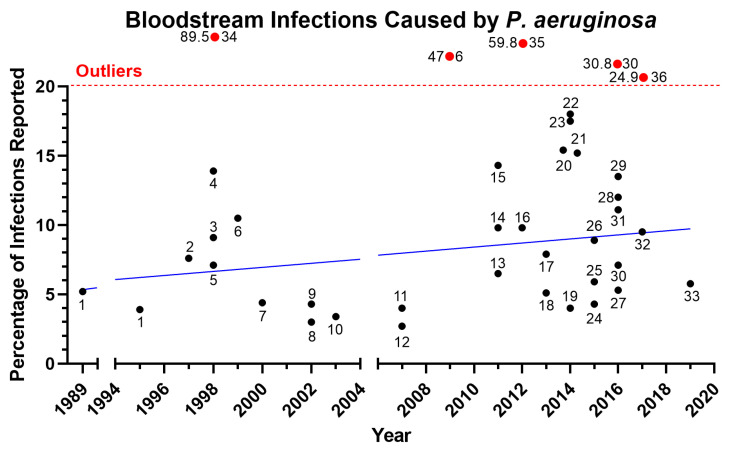
Rates for bloodstream infections caused by *P. aeruginosa* in hospitalized patients over the past 3 decades (1989–2019). Hospital-acquired infections included bloodstream infections and catheter-associated bloodstream infections, catheter-associated urinary tract infection, surgical site infections and burn wound infections, and hospital-associated and ventilator-associated pneumonia; although not every article reported every category, most included at least three. Each circle represents the rate reported in the study, and the year corresponds to the last collection date of the data. The solid blue line represents the linear regression of the infection rate over the years excluding outliers, which are shown above the dotted red line. The numbers associated with the black circles indicate reference numbers; the numbers to the left of the red circles are the percentages, and those to the right are the reference numbers. **References:** 1. [5], 2. [39], 3. [22], 4. [19], 5. [40], 6. [41], 7. [42], 8. [43], 9. [3], 10. [7], 11. [44], 12. [45], 13. [46], 14. [47], 15. [48], 16. [49], 17. [50], 18. [51], 19. [10], 20. [52], 21. [11], 22. [53], 23. [54], 24. [55], 25. [56], 26. [57], 27. [58], 28. [59], 29. [60], 30. [61], 31. [16], 32. [17], 33. [8]. Outliers: 34. [62], 35. [21], 36. [63].

**Table 1 microorganisms-11-00916-t001:** (**A**). Novel antibiotics against *P. aeruginosa. (***B**). Beta-lactamase inhibitors targeting *P. aeruginosa*.

**(A) Novel antibiotics against *P. aeruginosa.***						
**Treatment/Study**	**Mechanism of Action**	**Population**	**Purpose**	**Phase**	**Number of Patients**	**Year, Reference**
Inhaled fosfomycin/tobramycin (FTI) after aztreonam inhalation solution (AZLI) run-in (**NCT00794586**)	Protein synthesis inhibition	Cystic fibrosis patients with *P. aeruginosa* respiratory tract infections	Efficacy, safety, and ability to maintain forced expiratory volume (FEV1) in 1 s	II	120	2013, [188]
Fosfomycin IV (ZTI-01) and oral fosfomycin (**NCT02178254**)	Protein synthesis inhibition	Hospital volunteers	Pharmacokinetics, safety	I	30	2016, [190]
POL7080 (murepavadin) (**NCT02096315**)	Outer membrane protein inhibitor	Non–cystic fibrosis patients with bronchiectasis	Safety, pharmacokinetics, pharmacodynamics	II	20	2017, [191]
Murepavadin (**NCT02110459**)	Outer membrane protein inhibitor	Healthy volunteers and renal failure patients	Pharmacokinetics, tolerability, safety	I	32	2019, [192]
Murepavadin (PRISM-MDR) (**NCT03409679**)	Outer membrane protein inhibitor	Ventilated ICU patients	Treating *P. aeruginosa* VAP	III	41	2019, [193]
Murepavadin (PRISM-UDR) (**NCT03582007**)	Outer membrane protein inhibitor	Hospital patients	Treating *P. aeruginosa* hospital-associated pneumonia	III	2	2019, [194]
RC01 (**NCT03832517**)	LPS binding	Healthy adult volunteers	Pharmacokinetics, tolerability, safety	I	8	2019, [195]
Polymyxin B analogue SPR741 (**NCT03376529**)	Plasma membrane disruption and/or antibiotic sensitization	Healthy adult volunteers	Pharmacokinetics, tolerability, safety	I	27	2018, [196]
Polymyxin B analogue SPR741 (**NCT03022175**)	Plasma membrane disruption and/or antibiotic sensitization	Healthy adult volunteers	Pharmacokinetics, tolerability, safety	I	64	2017, [197]
Polymyxin B analogue SPR206 (**NCT03792308**)	Plasma membrane disruption and/or antibiotic sensitization	Healthy human volunteers	Pharmacokinetics, tolerability, safety	I	94	2020, [198]
Polymyxin B analogue SPR206 (**NCT03376529)**	Plasma membrane disruption and/or antibiotic sensitization	Healthy human volunteers	Pharmacokinetics, tolerability, safety	I	27	2018, [196]
Polymyxin B analogue MRX-8 (**NCT04649541**)	Plasma membrane disruption and/or antibiotic sensitization	Healthy human volunteers	Pharmacokinetics, tolerability, safety	I	116	2021, [199]
**(B) Beta-lactamase inhibitors targeting *P. aeruginosa.***						
**Treatment/Study**	**Mechanism of Action**	**Population**	**Purpose**	**Phase**	**Number of Patients**	**Year, Reference**
Imipenem/relebactam/cilastatin (**NCT02452047 RESTORE-IMI 1**)	Beta-lactamase inhibitor	Patients with HABP, VAPB, urinary tract infections, and intra-abdominal infections	Efficacy at treating HABP, VAPB, urinary tract infections, and intra-abdominal infections	III	31	2019, [200,201]
Imipenem/relebactam/cilastatin (**NCT02493764**-**RESTORE-IMI 2**)	Beta-lactamase inhibitor	Patients with HABP or VAPB	Efficacy at treating HABP or VAPB	III	264	2021, [202,203]
Imipenem/cilastatin/relebactam (**NCT05561764**)	Beta-lactamase inhibitor	Patients with HABP and VAPB	Efficacy at treating HABP and VAPB	III	274	2022, [204]
Imipenem/relebactam/Cilastatin **(NCT03583333)**	Beta-lactamase inhibitor	Patients with CF pneumonia	Efficacy at treating with CF pneumonia	IV	16	2023, [205]
Imipenem/cilastatin/XNW4107 (**NCT05204563**)	Beta-lactamase inhibitor	Patients with HABP and VAPB	Efficacy at treating patients with HABP and VAPB	III	450	2023, [206]
Nacubactam (**NCT02134834, NCT02972255,** and **NCT03182504**)	Beta-lactamase inhibitor	Healthy human volunteers	Pharmacokinetics, tolerability, safety	I	21	2018, [207,208,209]

**Table 2 microorganisms-11-00916-t002:** Bacteriophage therapy for *P. aeruginosa*.

Treatment/Study	Mechanism of Action	Population	Purpose	Phase	Number of Patients	Reference/Year
MUCOPHAGES (10 *Pseudomonas* specific bacteriophages) (**NCT01818206**)	Binding of bacteriophage to specific *Pseudomonas* targets to induce lysis	Cystic fibrosis sputa	Clinical study	--	59	2012, [225]
PhagoBurn (PP1131) 12 natural lytic *Pseudomonas* bacteriophages (**NCT02116010**)	Binding of bacteriophage to specific *Pseudomonas* targets to induce lysis	Burn patients	Treat *P. aeruginosa* burn wound infection	I/II	--	2018, [224]
B-PAO1 (4 *Pseudomonas*-specific bacteriophages) (**NCT03395743**)	Binding of bacteriophage to specific *Pseudomonas* targets to induce lysis	Cystic fibrosis patients	Prevent/treat *P. aeruginosa* infection	--	--	2019, [226]
Bacteriophage cocktail spray (Phage cocktail-SPK) (**NCT04323475**)	Binding of bacteriophage to specific *Pseudomonas* targets to induce lysis	Burn patients	Treatment of *P. aeruginosa* and infection of burn wounds	I	12	2023, [227]
AP-PA02 (**NCT04596319**)	Binding of bacteriophage to specific *Pseudomonas* targets to induce lysis	Chronic *P. aeruginosa* lung infections and CF	Treatment of chronic *P. aeruginosa* lung infections and CF	I/II	29	2022, [228]
YPT-01 (**NCT04684641**)	Binding of bacteriophage to specific *Pseudomonas* targets to induce lysis	Chronic *P. aeruginosa* lung infections and CF	Treatment of chronic *P. aeruginosa* lung infections and CF	I/II	8	2023, [229]

**Table 3 microorganisms-11-00916-t003:** Strategies targeting *P. aeruginosa* virulence.

Treatment/Study	Mechanism of Action	Population	Purpose	Phase	Number of Patients	Reference/Year
Azithromycin to inhibit QS (**NCT00610623**)	Biofilm formation inhibition	Ventilated patients	Reduced *P. aeruginosa* VAP	II	92	2018, [238]
OligoG (**NCT00970346, NCT03822455, and NCT03698448)**	Biofilm formation inhibition	Healthy volunteers	Pharmacokinetics, tolerability, safety	I	26	2022, [240,241]
PLG0206 peptide (**NCT05137314**)	Biofilm formation inhibition	Patients with prosthetic joints	Treatment for prosthetic joint infection	I	14	2022, [242]
Fluorothiazinon (aka Ftortiazinon) (**NCT03638830**)	Targeting the type III secretion system	Patients with *P. aeruginosa* complicated urinary tract infection	Adjunct treatment safety, efficacy	II	777	2021, [243]

**Table 4 microorganisms-11-00916-t004:** *P. aeruginosa* immunotherapy/monoclonal antibodies.

Treatment/Study	Population	Purpose	Phase	Number of Patients	Reference/Year
MEP IGIV (mucoid exopolysaccharide Ab) (**NCT00004747**)	Cystic fibrosis patients	Reduce frequency of acute pulmonary exacerbation and mucoid *P. aeruginosa* colonization	II	170	2005, [261]
KB001 (anti-Pa PcrV Fab Ab; humanized, PEGylated recombinant) (**NCT00691587**)	Ventilated patients	Prevent *P. aeruginosa* VAP	Pilot	36	2009, [262]
KB001 (anti-Pa PcrV Fab Ab; humanized, PEGylated recombinant) (**NCT00638365**)	Cystic fibrosis patients	Safety, tolerability, pharmacokinetics, pharmacodynamics, and prevent *P. aeruginosa* respiratory tract infection	I/II	27	2014, [263]
KB001 (anti-Pa PcrV Fab Ab; humanized, PEGylated recombinant) (**NCT01695343**)	Cystic fibrosis patients	Reduce *P. aeruginosa* infection	II	169	2018, [264]
MEDI3902 (anti-PcrV + anti-Psl Mab) (**NCT02255760**)	Healthy volunteers	Pharmacokinetics, safety	I	56	2019, [265]
MEDI3902 (anti-PcrV + anti-Psl Mab) (**NCT02696902**)	Ventilated patients	Prevent *P. aeruginosa* VAP	II	168	2020, [266]
PseudIgY (avian Ab to Pa) (**NCT00633191**)	Cystic fibrosis patients’ safety	Prevent *P. aeruginosa* respiratory tract colonization; preserve pulmonary function	I/II	14	2012, [267]
PseudIgY (avian Ab to Pa) (**NCT01455675**)	Cystic fibrosis patients	Reduce *P. aeruginosa* reinfection	III	164	2017, [268]
KBPA-101 (panobacumab) human IgM anti-Pa O11 LPS Mab obtained from a volunteer immunized with LPS-toxin A conjugate vaccine (**NCT00851435**)	Patients with *P. aeruginosa* hospital-acquired pneumonia	Safety, Pharmacokinetics, adjunct treatment of *P. aeruginosa* hospital-acquired pneumonia	I/II	14	2014, [269,270,271]
AR-105 (Aerucin, anti-alginate Mab) (**NCT03027609**)	Ventilated patients	Adjunct treatment of *P. aeruginosa* VAP	II	158	2019, [272]
Pentaglobin (PENTALLO) (**NCT03494959**)	Neutropenic acute leukemia or transplant patients with carbapenem-resistant *Enterobacteriaceae*	Decrease *P. aeruginosa* mortality	II	120	2023, [273]
TRL1068 monoclonal antibody (**NCT04763759**)	Prosthetic joint infection patients secondary to *P. aeruginosa*	Safety and pharmacokinetics. Decrease C-reactive protein (CRP), erythrocyte sedimentation rate (ESR), IL-6, and IL-10	I	18	2023, [274]

**Table 5 microorganisms-11-00916-t005:** *P. aeruginosa* outer membrane proteins as a vaccine.

Treatment/Study	Mechanism of Action	Population	Purpose	Phase	Number of Patients	Reference/Year
IC43 vaccine (recombinant *P. aeruginosa* outer membrane protein OprF/I) (**NCT00778388**)	Induce immune response against *P. aeruginosa* outer membrane protein (OprF/I)	Healthy volunteers	Safety pharmacokinetics immunogenicity	I	163	2013, [297]
IC43 vaccine (recombinant *P. aeruginosa* outer membrane protein OprF/I) (**NCT00876252**)	Induce immune response against *P. aeruginosa* outer membrane protein (OprF/I)	Ventilated patients	Safety immunogenicity	II	400	2017, [298]
IC43 vaccine (recombinant *P. aeruginosa* outer membrane protein OprF/I) (**NCT01563263**)	Induce immune response against *P. aeruginosa* outer membrane protein (OprF/I)	Ventilated ICU patients	Efficacy in preventing *P. aeruginosa* infection immunogenicity safety	II/III	800	2020, [299]

**Table 6 microorganisms-11-00916-t006:** Strategies targeting *P. aeruginosa* iron acquisition systems.

Treatment/Study	Mechanism of Action	Population	Purpose	Phase	Number of Patients	Reference/Year
Gallium (GANITE), inhaled (**NCT01093521**)	Decreasing bacterial Fe uptake and interfering with Fe signaling	Cystic fibrosis patients with chronic *P. aeruginosa* respiratory tract infection	Pharmacokinetics, safety, and reduce *P. aeruginosa* burden	I	20	2013, [157]
Gallium (IGNITE) (**NCT02354859**)	Decreasing bacterial Fe uptake and interfering with Fe signaling	Cystic fibrosis patients	Improve pulmonary function	II	119	2018, [300]
Inhaled sodium nitrite (**NCT02694393**)	Decreasing bacterial Fe uptake and interfering with Fe signaling	Cystic fibrosis patients with chronic *P. aeruginosa* infection	Safety and efficacy reduced *P. aeruginosa* bioburden	I/II	35	2023, [301]
Cefiderocol (**NCT03032380**)	Decreasing bacterial Fe uptake and interfering with Fe signaling	Nosocomial pneumonia secondary to *P. aeruginosa*	Efficacy against nosocomial pneumonia secondary to *P. aeruginosa*	III	300	2020, [303]

## Data Availability

Not applicable.

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
