# Peer review of "Anti-Pseudomonas aeruginosa Vaccines and Therapies: An Assessment of Clinical Trials"

_microorganisms, 2023, doi:10.3390/microorganisms11040916_

Round 1

Reviewer 1 Report

This is a significance work.  Discussing both vaccines and therapeutics in one paper would be helpful to conceptualize gaps especially given the significance of the topic.

A few comments:

1.  The literature review is substandard.  There are several studies in this space that are not discussed or referenced.  A more thorough lit review review is warranted, especially for molecular targets. There are several key factors missing both clinically and molecularly.

2. The organization of this paper is not clearly conveyed. There is an introduction that is followed by numbered paragraphs.  it is not clear what is part of intro and what the substance of the review. This extends throughout the paper. The authors transition from pseudomonas-related indications to vaccines without organizational identification of the transition (from section 4 to 5).

3. Section 5 is awkward.  It is important as it should set the stages for what is covered and not covered in treatments, but is not organized or presented  clearly. Further the analysis does not tie together  the gaps clearly.  A table that lines up indications, targets, and therapies/vacciens would tie this paper together more clearly.

4.  The first paragraph of the conclusion would be better delivered in small bite-size paragraph cover the key points that are laid out in the introduction.  

All of the material is here for a solid review. The organization isn't as clear as it could be. With some help there, this is an excellent review of the material.

Author Response

Reviewer 1

This is a significance work.  Discussing both vaccines and therapeutics in one paper would be helpful to conceptualize gaps especially given the significance of the topic.

A few comments:

  1. The literature review is substandard.  There are several studies in this space that are not discussed or referenced.  A more thorough lit review review is warranted, especially for molecular targets. There are several key factors missing both clinically and molecularly.

We appreciate the reviewer’s comment. We agree with the need for expanding the literature on this topic. We went ahead and double checked the literature for other clinical trials. Specifically, we added additional studies on polymyxin B derivatives, beta lactamase inhibitors, anti-biofilm agents, anti-microbial peptides, and iron chelating agents. We have also described the structure of particular molecules and their inclusion in different clinical trials.

  1. The organization of this paper is not clearly conveyed. There is an introduction that is followed by numbered paragraphs.  it is not clear what is part of intro and what the substance of the review. This extends throughout the paper. The authors transition from pseudomonas-related indications to vaccines without organizational identification of the transition (from section 4 to 5).

We appreciate the reviewer’s comment. We divided the manuscript using specific numbers and with regards to specific groups of anti-pseudomonal agents. At the end of the introduction, we described the general organization of the review. At end of the beginning of the clinical trial sections, we also provided a general summary of the different types of anti-pseudomonal agents that we will be describing. In addition, we added another transition after the epidemiology section on different Pseudomonal diseases to transition to the virulence factors. We also did this to transition from the virulence factors to the clinical trials section as well. Furthermore, we replaced figure 1 with three separate figures. These figures include citations for the studies used in the analysis. In addition, these figures describe hospital acquired infections, lower respiratory tract infections, and blood stream infections caused by P. aeruginosa. This provides deeper context to the epidemiology of the main diseases we address in this review.

  1. Section 5 is awkward.  It is important as it should set the stages for what is covered and not covered in treatments, but is not organized or presented  clearly. Further the analysis does not tie together  the gaps clearly.  A table that lines up indications, targets, and therapies/vacciens would tie this paper together more clearly.

We appreciate the reviewer’s comment. We added separate numbers to delineate different sections within each treatment discussed. We also made sure to expand the literature as mentioned before. In addition, we also added the targets of our analysis as well as the indications as previously mentioned. In addition, we provide backgrounds to different treatments on their mechanism and indicators before delving into the clinical trial data.

  1. The first paragraph of the conclusion would be better delivered in small bite-size paragraph cover the key points that are laid out in the introduction.  

We appreciate the reviewer’s comments. We have already separated the first paragraph of the conclusion into three small sized paragraphs.

All of the material is here for a solid review. The organization isn't as clear as it could be. With some help there, this is an excellent review of the material.

Reviewer 2 Report

I have read the manuscript entitled " Anti-Pseudomonas aeruginosa vaccines and therapies: An Assessment of Clinical Trials" with great interest and I think it is in principle suited for a publication in the Microorganisms, Special Issue “Identification and Characterization of Clinically Relevant Bacteria”. This is a very important biomedical area, and the review will be interesting for the readers of Microorganisms. However, I also have concerns and comments. The following corrections and additions should be made.

Comments:

Figure 1. I think that the results for Pseudomonas species and not specifically for P. aeruginosa are discussed in the papers ## 8, 12, and 23. Please check. In the reference #20 (Song et al.), abstract: “Pseudomonas aeruginosa was the most common (n=2997, 45.7%) isolate from the burn patients”, please check this percentage according with data in Figure 1.

Tables I-VII. “Reference/Year” What is "year"? May be “Study Start Date” or “Actual Study Completion Date” according to information from ClinicalTrials.gov? If this is the year of publication, then please check the correspondence between the data in the tables and the data of the year of publication in the References section. Listed below are examples.

Table I. “[120], 2010”. Please check the year in the References section for the reference #120 (2012?).

Table II. “[134], 2023”. (2021?).

Table III. “[143], 2018”. (2010?).

Table IV. “[152], 2012”. (2016?).

Table V. “[174], 2013”. (2014?).

Table VII. “[186], 2013”. (2018?).

Lines 53-55. “In immunocompetent patients, P. aeruginosa infection  is primarily linked to the use of central venous catheters, traumatic or surgical wound infections, or indwelling urinary catheters [37].” Please check the relevance of the reference #37.

Lines 104-107. “Mutations in CFTR, as seen in CF patients, cause a dry and thick mucus discharges that effect multiple organ systems, including the hepatobiliary system, pancreas, gastrointestinal tract, and male reproductive tract [61,62].” I think that mutations in CFTR are not discussed in the cited works ##61, 62. Please, check.

Lines 113-116. “Additional infections with built-in resistance to numerous antibiotics, such as Stenotrophomonas maltophila, Achromobacter xyloxidans, Burkholderia cepacia complex, nontuberculous mycobacteria, and others, may predominate with advancing age or illness [64,65].” I think that the microorganisms (Stenotrophomonas maltophila, Achromobacter xyloxidans, Burkholderia cepacia) are not discussed in the cited works ##64, 65. Please, check.

Lines 117-119. “The incidence of P. aeruginosa infection in CF patients increases from 20% in individuals under the age of five to as high as 70% by the age of 18 [64,65].” In the cited work #65, in Introduction: “Epidemiological data from the Cystic Fibrosis Foundation, USA, that includes more than 24,000 patients, revealed that approximately 27% of patients aged between 2 and 5 years, rising to approximately 80% of those between the ages of 25 and 34 years, have chronic P. aeruginosa infection”. Please check and correct.

Lines 155-157. “Pseudomonas aeruginosa accounts for 57% of the bacterial pathogens obtained from positive swab and tissue culture…” Please add relevant reference.

Lines 159-160. Pseudomonas aeruginosa colonization at the burn eschar site may produce up to 105 bacteria per gram of tissue [74-76]. Please check that this statement is observed in all references #74-76. If it isn't, please correct.

Lines 162-164. Following such invasion, microorganisms may multiply in necrotic tissue and enter blood vessels leading to the development of secondary infection [74-76]. Please check the relevance of these references.

Line 178. “LPS,…” please describe the first mentioned abbreviation.

Lines 181-182. I think that pqs QS system are not discussed in the cited works ##82-84. Please check.

Lines 204-205. “It also enhances the elastolytic activity of LasB [102]. I think that LasB is not discussed in the cited work #102. Please check.

Lines 226-227. “WaaL (necessary for LPS biosynthesis) [116-118]”. I think that “WaaL” is not discussed in the cited works ##116-118. Please check.

Lines 269-270, 288, 339, 412, 485-486, 578-579, 667, 693. “The results of the test have not been published.” I think that this information should be indicated, if the authors are absolutely sure that the results of the test have not been published, if it isn't, then this assumption is better removed from the review.

Lines 295-297. “The structure of PG-1 contains six positively charge arginine residues and four positively charged cysteine residues that form two antiparallel β-sheets with a β-turn [128].” I think that the structure of PG-1 is not discussed in the cited work #128. Please check.

Lines 418-420. “Specifically, azithromycin inhibits the 23S rRNA of the 50S ribosome unit, which decreased the production of genes essential for QS functions in P. aeruginosa [139-142]. I think that this is not discussed in all the cited papers.

Lines 449-450. “the immunogen was used to elicit opsonizing anti-MEP antibodies in humans 449 [160]”. I think that a study in rats observed in the cited article #160. Please check.

Lines 552-553. “The PseudIgY and the control groups were told to gargle and ingest either solution [153]”. I think that “PsAer-IgY” is correct; “PseudIgY” is not discussed in [153].

Lines 647-648. “Numerous studies using both the murine models as well as the ex vivo model 647 demonstrated the role of the T3SS in the virulence of P. aeruginosa [108,162,177-181].” I think that “T3SS” are not discussed in [181]. Please check.

Lines 695-770. Conclusion.

It is unclear why the authors do not consider antimicrobial peptides as perspective therapy against P. aeruginosa (e.g. https://doi.org/10.3389/fmicb.2019.03053 , https://doi.org/10.1016/j.ejmech.2019.111814 , https://doi.org/10.3390/ijms22189776 , https://doi.org/10.1038/s41598-019-42440-2, https://doi.org/10.3390/antibiotics12020389).

Best regards.

Author Response

Reviewer 2

I have read the manuscript entitled " Anti-Pseudomonas aeruginosa vaccines and therapies: An Assessment of Clinical Trials" with great interest and I think it is in principle suited for a publication in the Microorganisms, Special Issue “Identification and Characterization of Clinically Relevant Bacteria”. This is a very important biomedical area, and the review will be interesting for the readers of Microorganisms. However, I also have concerns and comments. The following corrections and additions should be made.

Comments:

Figure 1. I think that the results for Pseudomonas species and not specifically for P. aeruginosa are discussed in the papers ## 8, 12, and 23. Please check. In the reference #20 (Song et al.), abstract: “Pseudomonas aeruginosa was the most common (n=2997, 45.7%) isolate from the burn patients”, please check this percentage according with data in Figure 1.

-We appreciate the reviewer’s comment. We replaced figure 1 with three separate figures. These figures include citations for the studies used in the analysis. In addition, these figures describe hospital acquired infections, lower respiratory tract infections, and blood stream infections caused by P. aeruginosa.

Tables I-VII. “Reference/Year” What is "year"? May be “Study Start Date” or “Actual Study Completion Date” according to information from ClinicalTrials.gov? If this is the year of publication, then please check the correspondence between the data in the tables and the data of the year of publication in the References section. Listed below are examples

-We appreciate the reviewer’s suggestion. We changed the reference year for trials listed on clinicaltrials.org to be when the study was completed. Any study that was published was referenced also by it’s publication year.

Table I. “[120], 2010”. Please check the year in the References section for the reference #120 (2012?).

-We appreciate the reviewer’s comment. We double checked and edited the years of the clinical trials mentioned in the tables to match the year completed on clinicaltrials.org or the year of the published publication.

Table II. “[134], 2023”. (2021?).

Table III. “[143], 2018”. (2010?).

Table IV. “[152], 2012”. (2016?).

Table V. “[174], 2013”. (2014?).

Table VII. “[186], 2013”. (2018?).

Lines 53-55. “In immunocompetent patients, P. aeruginosa infection  is primarily linked to the use of central venous catheters, traumatic or surgical wound infections, or indwelling urinary catheters [37].” Please check the relevance of the reference #37.

-We appreciate the reviewer’s comment. We removed this section from the review.

Lines 104-107. “Mutations in CFTR, as seen in CF patients, cause a dry and thick mucus discharges that effect multiple organ systems, including the hepatobiliary system, pancreas, gastrointestinal tract, and male reproductive tract [61,62].” I think that mutations in CFTR are not discussed in the cited works ##61, 62. Please, check.

-We appreciate the reviewer’s comment. We removed these citations from the review and replaced them with more appreciate references (78,79).

Lines 113-116. “Additional infections with built-in resistance to numerous antibiotics, such as Stenotrophomonas maltophila, Achromobacter xyloxidans, Burkholderia cepacia complex, nontuberculous mycobacteria, and others, may predominate with advancing age or illness [64,65].” I think that the microorganisms (Stenotrophomonas maltophila, Achromobacter xyloxidans, Burkholderia cepacia) are not discussed in the cited works ##64, 65. Please, check.

-We appreciate the reviewer’s comment. We removed these citations from the review and replaced them with more appreciate references (81,82).

Lines 117-119. “The incidence of P. aeruginosa infection in CF patients increases from 20% in individuals under the age of five to as high as 70% by the age of 18 [64,65].” In the cited work #65, in Introduction: “Epidemiological data from the Cystic Fibrosis Foundation, USA, that includes more than 24,000 patients, revealed that approximately 27% of patients aged between 2 and 5 years, rising to approximately 80% of those between the ages of 25 and 34 years, have chronic P. aeruginosa infection”. Please check and correct.

-We appreciate the reviewer’s comment. We removed these citations from the review and replaced them with more appreciate references. We used the Cystic Fibrosis Foundation Patient Registry 2020 Annual Report.

Lines 155-157. “Pseudomonas aeruginosa accounts for 57% of the bacterial pathogens obtained from positive swab and tissue culture…” Please add relevant reference.

-We appreciate the reviewer’s comment. We removed this section from the review.

Lines 159-160. Pseudomonas aeruginosa colonization at the burn eschar site may produce up to 105 bacteria per gram of tissue [74-76]. Please check that this statement is observed in all references #74-76. If it isn't, please correct.

-We appreciate the reviewer’s comment. We removed this section from the review.

Lines 162-164. Following such invasion, microorganisms may multiply in necrotic tissue and enter blood vessels leading to the development of secondary infection [74-76]. Please check the relevance of these references.

-We appreciate the reviewer’s comment. We removed this section from the review.

Line 178. “LPS,…” please describe the first mentioned abbreviation.

-We appreciate the reviewer’s comment. We wrote down the full name of LPS.

Lines 181-182. I think that pqs QS system are not discussed in the cited works ##82-84. Please check.

-We appreciate the reviewer’s comment. We checked reference 82 and it does describe the pqs system (97). 

Lines 204-205. “It also enhances the elastolytic activity of LasB [102]. I think that LasB is not discussed in the cited work #102. Please check.

-We appreciate the reviewer’s comment. We added another reference to this section as suggested and moved another reference to its appropriate location.

Lines 226-227. “WaaL (necessary for LPS biosynthesis) [116-118]”. I think that “WaaL” is not discussed in the cited works ##116-118. Please check.

-We appreciate the reviewer’s comment. We added another reference to describe the synthesis of LPS (135).

Lines 269-270, 288, 339, 412, 485-486, 578-579, 667, 693. “The results of the test have not been published.” I think that this information should be indicated, if the authors are absolutely sure that the results of the test have not been published, if it isn't, then this assumption is better removed from the review.

-We appreciate the reviewer’s comment. We removed all references to anything about the status of the clinical trials.

Lines 295-297. “The structure of PG-1 contains six positively charge arginine residues and four positively charged cysteine residues that form two antiparallel β-sheets with a β-turn [128].” I think that the structure of PG-1 is not discussed in the cited work #128. Please check.

-We appreciate the reviewer’s comment. We added the following references (151,152).

Lines 418-420. “Specifically, azithromycin inhibits the 23S rRNA of the 50S ribosome unit, which decreased the production of genes essential for QS functions in P. aeruginosa [139-142]. I think that this is not discussed in all the cited papers.

-We appreciate the reviewer’s comment. We removed this statement from the review. The references included describe the full effect of azithromycin on QS.

Lines 449-450. “the immunogen was used to elicit opsonizing anti-MEP antibodies in humans 449 [160]”. I think that a study in rats observed in the cited article #160. Please check.

-We appreciate the reviewer’s comment. We changed the word human to rat.

Lines 552-553. “The PseudIgY and the control groups were told to gargle and ingest either solution [153]”. I think that “PsAer-IgY” is correct; “PseudIgY” is not discussed in [153].

-We appreciate the reviewer’s comments. We changed our original abbreviation to the suggested abbreviation PsAer-IgY

Lines 647-648. “Numerous studies using both the murine models as well as the ex vivo model 647 demonstrated the role of the T3SS in the virulence of P. aeruginosa [108,162,177-181].” I think that “T3SS” are not discussed in [181]. Please check.

-We appreciate the reviewer’s comment. We added the reference from Hauser et al (204), which is a comprehensive description of the P. aeruginosa

Lines 695-770. Conclusion.

It is unclear why the authors do not consider antimicrobial peptides as perspective therapy against P. aeruginosa (e.g. https://doi.org/10.3389/fmicb.2019.03053 , https://doi.org/10.1016/j.ejmech.2019.111814 , https://doi.org/10.3390/ijms22189776 , https://doi.org/10.1038/s41598-019-42440-2, https://doi.org/10.3390/antibiotics12020389).

-We appreciate the reviewer’s comment. We double checked the literature and found one clinical trial using anti-microbial peptides for P. aeruginosa. Therefore, we did not include this in the conclusion. Instead, we made a separate section (3.3.3 Anti-microbial Peptides) in the clinical trial portion that included the below papers and the clinical trial being investigated using this peptide PLG0206 (WLBU2).

Round 2

Reviewer 2 Report

The manuscript entitled "Anti-Pseudomonas aeruginosa vaccines and therapies: An Assessment of Clinical Trials” has been significantly improved and I can suggest that manuscript can be published after minor text editing.

Minor comments:

Line 71. “LTRIs” – LRTI is more accurate abbreviation.

Figure 1. Please check the percentage for dots 4 according Figure 1 in the reference #7. Please check two dots 6 for 2014 year. 

Figure 2. Please check the blue square 10 for 2015 year.

Figure 3. Also please check two dots 24 for 2015 year (in the cited work #55 are demonstrated only 4.9%).

Tables IA, IB, II, III, IV, V, VII. If the column heading is "Year of Completion, Reference", then "Year of Completion" must be indicated first in the corresponding row. For example, in table IA it should be: 2013 [137] instead of "[137], 2013". Please correct in all tables. 

Table VII – Please check the table numbering, I think it's Table VI.

Line 605. “Anti-microbial Peptides” - Antimicrobial Peptides.

Line 613. “Staph aureus” - Staphylococcus aureus.

Best regards.

Author Response

The manuscript entitled "Anti-Pseudomonas aeruginosa vaccines and therapies: An Assessment of Clinical Trials” has been significantly improved and I can suggest that manuscript can be published after minor text editing.

Minor comments:

Line 71. “LTRIs” – LRTI is more accurate abbreviation.

We appreicate the reviewer's comment. We changed the abbreviation

Figure 1. Please check the percentage for dots 4 according Figure 1 in the reference #7. Please check two dots 6 for 2014 year. 

-We appreciate the reviewer's comment. We double checked and made changes to the references and data number

Figure 2. Please check the blue square 10 for 2015 year.

-We appreciate the reviewer's comment. We double checked and made changes to the references and data number

Figure 3. Also please check two dots 24 for 2015 year (in the cited work #55 are demonstrated only 4.9%).

-We appreciate the reviewer's comment. We double checked and made changes to the references and data number

Tables IA, IB, II, III, IV, V, VII. If the column heading is "Year of Completion, Reference", then "Year of Completion" must be indicated first in the corresponding row. For example, in table IA it should be: 2013 [137] instead of "[137], 2013". Please correct in all tables. 

-We appreciate the reviewer's comment. We made the changes to the table

Table VII – Please check the table numbering, I think it's Table VI.

-We appreciate the reviewer's comment. We made the changes to the paper

Line 605. “Anti-microbial Peptides” - Antimicrobial Peptides.

-We appreciate the reviewer's comment. We made the changes to the paper

Line 613. “Staph aureus” - Staphylococcus aureus.

-We appreciate the reviewer's comment. We made the changes to the paper